# Maintenance of neurotransmitter identity by Hox proteins through a homeostatic mechanism

Weidong Feng[1,2,3], Honorine Destain [1,2,3], Jayson J. Smith [1,2] & Paschalis Kratsios [1,2,3] ✉

Hox transcription factors play fundamental roles during early patterning, but they are also expressed continuously, from embryonic stages through adulthood, in the nervous system. However, the functional significance of their sustained expression remains unclear. In *C. elegans* motor neurons (MNs), we find that LIN-39 (Scr/Dfd/Hox4-5) is continuously required during post-embryonic life to maintain neurotransmitter identity, a core element of neuronal function. LIN-39 acts directly to co-regulate genes that define cholinergic identity (e.g., *unc-17/VAChT, cho-1/ChT*). We further show that LIN-39, MAB-5 (Antp/Hox6-8) and the transcription factor UNC-3 (Collier/Ebf) operate in a positive feedforward loop to ensure continuous and robust expression of cholinergic identity genes. Finally, we identify a two-component design principle for homeostatic control of Hox gene expression in adult MNs: Hox transcriptional autoregulation is counterbalanced by negative UNC-3 feedback. These findings uncover a noncanonical role for Hox proteins during post-embryonic life, critically broadening their functional repertoire from early patterning to the control of neurotransmitter identity.

Information flow in the nervous system critically relies on the ability of distinct neuron types to synthesize and package into synaptic vesicles specific chemical substances, known as neurotransmitters (NTs). Hence, a core functional feature of each neuron type is its NT identity, defined by the co-expression of genes encoding proteins necessary for the synthesis, packaging, and breakdown of a particular NT. Although rare instances of NT identity switching have been described in the nervous system[1,2], it is generally accepted that individual neuron types acquire a specific NT identity during development and maintain it throughout life.

In the case of cholinergic neurons, the enzyme choline acetyltransferase (ChAT) synthesizes acetylcholine (ACh) from its precursor choline, the vesicular ACh transporter (VAChT) packages ACh into synaptic vesicles, and the enzyme acetylcholinesterase (AChE) breaks down ACh upon its release into choline. The choline transporter (ChT) reuptakes choline back into the cholinergic neuron (Fig. 1a)[3]. Co-

expression of all these proteins throughout the life of a cholinergic neuron ensures the continuation of ACh biosynthesis, and thereby the communication of cholinergic neurons with their post-synaptic targets. Importantly, reduced expression of ACh pathway proteins has been associated with multiple neurological disorders[4–6]. Despite the widespread use of ACh in every nervous system[7,8], how the co-expression of ACh pathway proteins is controlled over time, from development through adulthood, is poorly understood. A handful of studies in *C. elegans* and mice identified LIM and POU homeodomain transcription factors as necessary during development for ACh pathway gene expression in various types of cholinergic neurons[9–12]. However, the mechanisms that maintain the expression of ACh pathway genes during post-embryonic life are largely unknown.

Motor neurons (MNs) in the spinal cord of vertebrates and the ventral nerve cord of many invertebrates use ACh to communicate with their muscle targets. The nematode *C. elegans* contains six

[1]Department of Neurobiology, University of Chicago, Chicago, IL, USA. [2]University of Chicago Neuroscience Institute, Chicago, IL, USA. [3]Committee on Development, Regeneration, and Stem Cell Biology, University of Chicago, Chicago, IL, USA. ✉e-mail: pkratsios@uchicago.edu

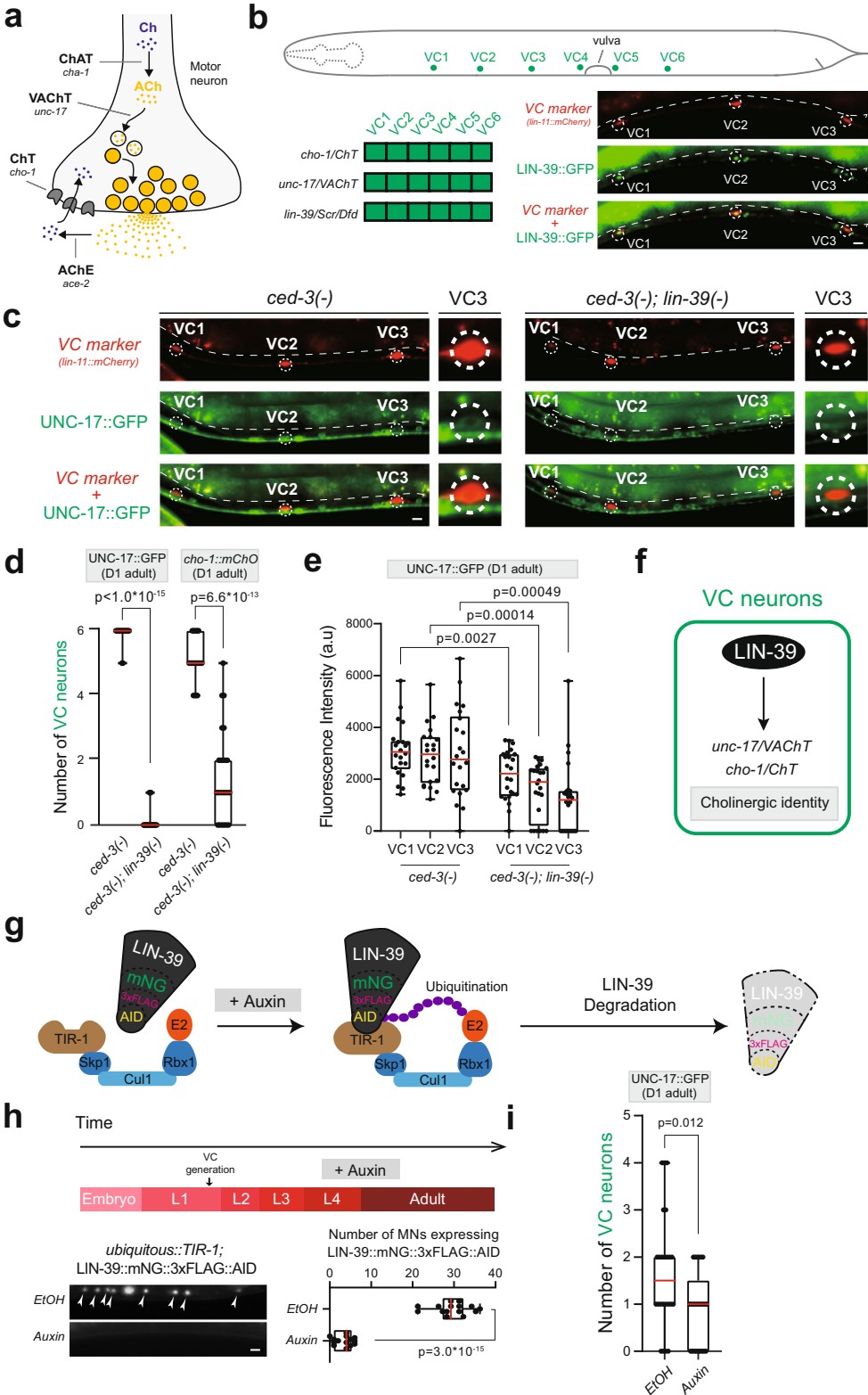

different classes (types) of MNs within its ventral nerve cord that are cholinergic[13]. Five classes (DA, DB, VA, VB, AS) control locomotion and are sex-shared (found in *C. elegans* males and hermaphrodites), whereas one class (VC) controls egg-laying and is found only in hermaphrodites. Each MN class contains several members (DA = 9 neurons, DB = 7, VA = 12, VB = 11, AS = 11, VC = 6) that intermingle along the nerve cord. The cholinergic identity of these MNs is defined by the expression of *unc-17* (VAChT), *cha-1* (ChAT), *ace-2* (AChE), and *cho-1* (ChT) (Fig. 1a)[14,15]. The rapid life cycle of *C. elegans* (~3 days from embryo to adult), the established methods to inactivate transcription factor activity in the adult, and the availability of reporter animals for all ACh pathway genes offer a unique opportunity to probe in vivo, and with single-cell resolution, the molecular mechanisms that maintain cholinergic identity during post-embryonic life.

**Fig. 1 | The Hox gene _lin-39 (Scr/Dfd/Hox4-5)_ is required to maintain the cholinergic identity of VC motor neurons. a** Presynaptic bouton of a cholinergic motor neuron. Acetylcholine (ACh) pathway genes: _cha-1/ChAT_: Choline Acetyltransferase, _unc-17/VAChT_: Vesicular Acetylcholine Transporter, _cho-1/ChT_: Choline Transporter, _ace-2/AChE_: Acetylcholinesterase. Schematic adapted from Kratsios, P., et al., Coordinated regulation of cholinergic motor neuron traits through a conserved terminal selector gene. _Nat. Neurosci._ 15(2), 205–214 (2012). **b** Cell body position of VC neurons in adult _C. elegans_. Bottom left: Co-expression of _cho-1, unc-17_ and _lin-39_ in VC motor neurons. Bottom right: Representative image of a fosmid-based _lin-39_ reporter (LIN-39::GFP) and a VC-specific marker (_lin-11::mCherry_). GFP and _mCherry_ co-localization occurs in all VC neurons; VC1-3 neurons are shown. **c** Representative images of UNC-17::GFP co-expressed with _lin-11::mCherry_ in _ced-3 (n1286)_ single and _ced-3 (n1286); lin-39 (n1760)_ double mutants (day 1 adults). Inset shows VC3. Dashed circles: cell body positions. Dashed white lines: intestinal boundary; intestine is autofluorescent in the green channel. **d** Quantification of _unc-17_ and _cho-1_ fosmid-based reporters in _ced-3 (n1286)_ single and _ced-3 (n1286); lin-39 (n1760)_ double mutants (day 1 adults). _n_ = 17–23 animals. **e** Fluorescence intensity quantification of UNC-17::GFP in VC1-3 in _ced-3 (n1286)_ single and _ced-3 (n1286); lin-39 (n1760)_ double mutants. See Material and Methods for details (a.u., arbitrary units). _n_ = 22 animals. **f** Summary of LIN-39 function in VC MNs. **g** The auxin-inducible protein degradation system. Skp1, Cul1, Rbx1, E2: components of E3 ligase complex. Worms express ubiquitously TIR1, a plant-specific substrate-recognizing subunit of the E3 ligase complex. With auxin, TIR1 binds to AID, leading to proteasomal degradation of LIN-39::mNG::3xFLAG::AID. **h** LIN-39::mNG::3xFLAG::AID depletion upon auxin. Representative images showing LIN-39::mNG::3xFLAG::AID expression in MNs upon auxin or EtOH treatment; quantifications on the right. _n_ = 14 animals. **i** Quantification of the number of VC neurons expressing UNC-17::GFP upon auxin or EtOH treatment. _n_ = 24 animals. For quantification in **d**, **e**, **h** and **i**, box and whisker plots were used with the presentation of all data points. Box boundaries indicate the 25th and 75th percentile. The limits indicate minima and maxima values, whereas center values (mean) are highlighted in red. An unpaired _t_-test (two-sided) with Welch's correction was performed. Source data are provided as a Source data file. Scale bars: 5 μm.

Members of the HOX family of homeodomain transcription factors play fundamental roles during early development in anterior–posterior patterning and body plan formation[16,17]. Seminal studies in worms, flies, zebrafish, and mice have also established critical roles for Hox genes during the early development of the nervous system[18-22]. For example, Hox gene inactivation can affect the specification and/or survival of neuronal progenitors, as well as the differentiation, survival, migration, and/or connectivity of young postmitotic neurons[18-22]. The focus on early development, however, combined with a lack of temporally controlled gene inactivation approaches that bypass early Hox pleiotropies (e.g., lethality, effects on neural progenitors) has resulted in an incomplete understanding of Hox gene functions in the nervous system. Emerging evidence in _C. elegans, Drosophila_, mice, and humans shows that Hox genes are expressed continuously, from development throughout adulthood, in certain neuron types[23-29]. However, the functional significance of sustained Hox gene expression in adult post-mitotic neurons remains unknown.

Individual neuron types acquire their NT identity during development and maintain it throughout life. Previous work in _C. elegans_ showed that the Hox gene _lin-39_ (Scr/Dfd/Hox4-5) is required for the production of serotonergic neurons in males and egg-laying (VC class) motor neurons in hermaphrodites[30-34], but whether _lin-39_ acts in postmitotic neurons to regulate NT identity genes was not tested. Hence, the role of Hox proteins in the control of NT identity remains unclear. Here, we show that LIN-39 is continuously required to maintain the expression of ACh pathway genes in cholinergic MNs, thereby securing their NT identity and function. In MNs that control locomotion, LIN-39 cooperates with another Hox protein MAB-5 (Antp/Hox6-8), and the transcription factor UNC-3 (Collier/Ebf) to directly activate the expression of ACh pathway genes. Moreover, LIN-39 and MAB-5 also regulate the expression levels of _unc-3_, thereby generating a positive feedforward loop that ensures the robustness of ACh pathway gene expression throughout life. Importantly, we find that Hox gene expression is under homeostatic control and propose a two-component design principle that maintains optimal levels of Hox gene expression in MNs over time: Hox transcriptional autoregulation (component 1) is counterbalanced by negative UNC-3 feedback (component 2). Because Hox genes are expressed in the adult nervous system of invertebrate and vertebrate animals, their functional role in the maintenance of NT identity may be deeply conserved.

## Results

### The Hox gene _lin-39_ maintains the identity of motor neurons necessary for egg laying

We initially focused on the hermaphrodite-specific VC MNs, for which regulators of their cholinergic identity remain unknown. The Hox gene _lin-39_ (Scr/Dfd/Hox4-5) is continuously expressed in VC neurons, from the time they are born (larval stage 1, L1) until adulthood (Fig. 1b) (Supplementary Fig. 1)[35]. However, animals carrying strong loss-of-function alleles for _lin-39_ lack VC neurons (_lin-39_ controls VC survival)[31,33], preventing us from testing its role in the regulation of VC cholinergic identity determinants _unc-17/VAChT_ and _cho-1/ChT_ (Supplementary Fig. 1)[15]. To bypass this, we generated double mutant animals for _lin-39_ and _ced-3_ (VC cell death depends on _ced-3_ caspase activity)[31,33], and indeed observed that VC neurons are normally generated in these animals (Fig. 1c, Supplementary Fig. 1d, e). Compared to controls, the number of VC neurons expressing _unc-17_ and _cho-1_ reporters was significantly reduced in _lin-39(n1760); ced-3(n1286)_ double mutants in the adult (day 1) stage (Fig. 1c, d). Subsequent fluorescence intensity analysis with higher resolution revealed that expression of the cholinergic marker _unc-17::gfp_ is significantly reduced, but not completely eliminated (Fig. 1e). For the _unc-17_ and _cho-1_ expression analysis, we used fosmid (~30 kb-long genomic clone)-based reporters, which despite their multicopy nature (overexpression), tend to faithfully recapitulate endogenous gene expression patterns[15]. Altogether, these findings suggest that, during larval development, _lin-39_ controls the expression of VC cholinergic identity determinants (Fig. 1f).

The sustained expression of _lin-39_ in VC MNs during adulthood raises the question of whether it is continuously required to maintain cholinergic identity. Because the _lin-39(n1760)_ allele removes gene activity starting in early development, we used a previously validated _lin-39::mNG::3xFLAG::AID_ allele that codes for a LIN-39 protein fused to mNG::3xFLAG (serves as an endogenous reporter) and auxin-inducible degron (AID) (enables LIN-39 depletion in a temporally-controlled manner)[36,37] (Fig. 1g). We initiated auxin treatment at the last larval stage (L4) and maintained animals on auxin until the first day of adulthood (Fig. 1h). This led to efficient LIN-39 depletion in VC neurons and other nerve cord MNs that normally express _lin-39_ (Fig. 1h). In day 1 adults, although we observed a hypomorphic effect on _unc-17_ expression in the control group (ethanol [EtOH] treatment), we did witness a statistically significant decrease in the number of VC neurons expressing the _unc-17_ reporter in animals treated with auxin (Fig. 1i). This indicates that _lin-39_ is required to maintain the expression of _unc-17/VAChT_ at later life stages. Lastly, we examined two additional markers of VC terminal differentiation, _ida-1_ (ortholog of human protein tyrosine phosphatase receptor type N [PTPRN])[38] and _glr-5_ (ortholog of human glutamate ionotropic receptor GRIK1)[39]. We found that _lin-39_ is continuously required to maintain _ida-1_ and _glr-5_ expression during late larval and adult stages (Supplementary Fig. 1d–f). In conclusion, _lin-39_, in addition to promoting VC survival during the L1 stage, is continuously required at later stages of life to control cholinergic identity gene expression (e.g., _unc-17_) and other features (_ida-1, glr-5_) of VC terminal differentiation (Fig. 1f).

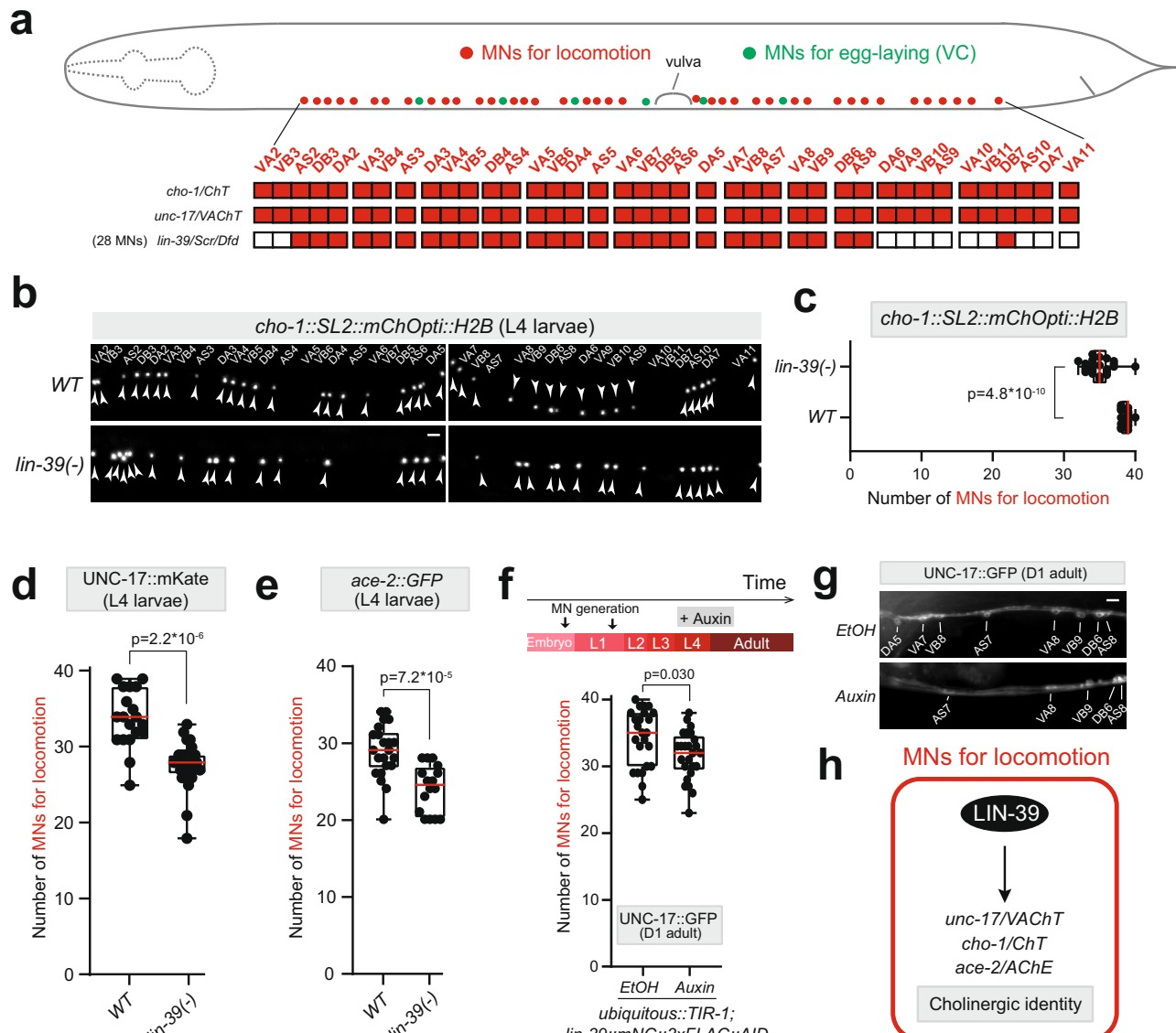

**Fig. 2 | The Hox gene *lin-39 (Scr/Dfd/Hox4-5)* controls the cholinergic identity of motor neurons necessary for locomotion. a** Schematic of a *C. elegans* hermaphrodite showing the cell bodies of MNs that control locomotion (red) and egg-laying (green). Expression profile of *cho-1, unc-17,* and *lin-39* is summarized in the color-filled table. **b** Representative images of a *cho-1* fosmid-based reporter in *WT* and *lin-39(n1760)* mutant animals (L4 stage). Motor neurons that control locomotion (e.g., DA, DB, VA, VB) express *cho-1* at L4. VC motor neurons mature later (day 1 adult) and do not express *cho-1* at L4. The identity of individual motor neurons is annotated. White arrowheads indicate MN nuclei because the *cho-1* reporter is localized to the nucleus (H2B). *mChOpti* signal is shown in white for better contrast. Left image: anterior to the vulva, Right image: posterior to the vulva. **c** Quantification of the number of motor neurons expressing *cho-1* in panel (**b**). The motor neurons that consistently lose *cho-1* expression in *lin-39(n1760)* mutants are

VA5, VB6, AS6, VA7, and VB8. *n* = 22 animals. **d** and **e** Quantification of the number of sex-shared motor neurons expressing *unc-17* (endogenous reporter *ot907* [UNC-17::mKate]) and *ace-2* (fosmid-based reporter *otEx4432* [*ace-2::GFP*]) in WT and *lin-39(n1760)* mutant hermaphrodites at L4. *n* = 19-25 animals. **f** and **g** Auxin treatment was initiated at L4 and images were scored at young adult stage (Day 1). Quantification of the number of sex-shared motor neurons expressing UNC-17::GFP upon auxin or EtOH treatment. *n* = 24 animals. For quantifications in **c**–**f**, box and whisker plots were used with the presentation of all data points. Unpaired *t*-test (two-sided) with Welch's correction was performed and *p*-values were annotated. Box boundaries indicate the 25th and 75th percentile. The limits indicate minima and maxima values, whereas center values (mean) are highlighted in red. **h** Schematic summarizing our findings in MNs that control locomotion. Source data are provided as a Source data file. Scale bars: 5 μm.

## *lin-39* controls the identity of motor neurons necessary for locomotion

Prompted by our findings in MNs that control egg-laying, we next asked whether the control of NT identity by *lin-39* extends to MNs that control locomotion. Using the endogenous *lin-39::mNG::3xFLAG::AID* reporter allele, we found *lin-39* to be continuously expressed, from development to adulthood, in 28 of the 39 cholinergic MNs necessary for locomotion (Fig. 2a) (Supplementary Fig. 1b)[35]. These neurons survive in animals carrying the strong loss-of-function *lin-39* allele *(n1760)* used in Fig. 1[40], enabling

assessment of putative *lin-39* effects on their cholinergic identity. Indeed, we found a significant decrease in the number of MNs expressing *cho-1/ChT* at L4 (Fig. 2b, c). We obtained similar results for two additional cholinergic identity markers using an endogenous reporter for *unc-17 (unc-17::mKate)* and a fosmid-based reporter for *ace-2/AChE* (Fig. 2d, e), suggesting *lin-39* co-regulates the expression of several ACh pathway genes in MNs necessary for locomotion. The observed reduction in ACh pathway gene expression may account for the previously described locomotion defects of *lin-39* mutant animals[36].

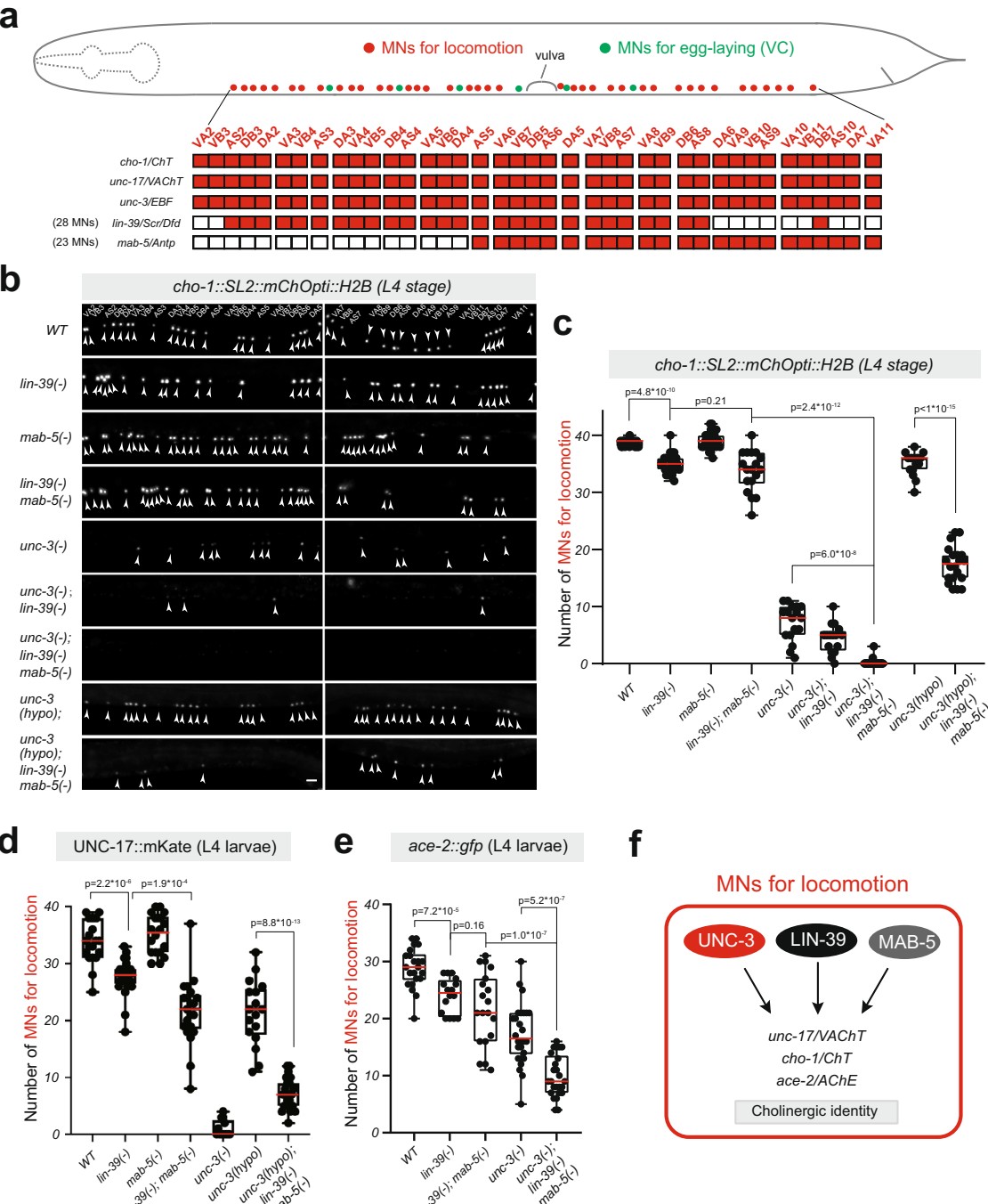

**Fig. 3 | *lin-39* cooperates with *mab-5 (Antp, Hox6-8)* and *unc-3 (Collier/Ebf)* to control cholinergic motor neuron identity. a** Schematic of a *C. elegans* hermaphrodite showing the cell bodies of MNs that control locomotion (red) and egg-laying (green). The expression profile of *cho-1, unc-17, unc-3, lin-39,* and *mab-5* is illustrated in the color-filled table below. **b** Representative images of a *cho-1* fosmid-based reporter at L4 in the genetic background of *WT, lin-39(-), mab-5(-), lin-39(-) mab-5(-), unc-3(-), unc-3(-); lin-39(-), unc-3(-); lin-39(-) mab-5(-), unc-3(hypo), and unc-3(hypo); lin-39(-) mab-5(-)*. Strong-loss-of-function (null) alleles of *lin-39(n1760), mab-5(e1239),* and *unc-3(n3435)* are annotated as *(-)* and the hypomorphic *unc-3* allele *ot837 [unc-3::mNG::AID]* as *(hypo)*. The identity of individual motor neurons is annotated. White arrowheads indicate MN nuclei because the *cho-1* reporter is localized to the nucleus (H2B). *mChOpti* signal is shown in white for better contrast.

Left image: anterior to the vulva, Right image: posterior to the vulva.
**c** Quantification of the number of motor neurons expressing *cho-1* in panel (**b**). *n* = 16–23 animals. **d** and **e** Quantification of the number of motor neurons (that control locomotion) expressing *unc-17* (endogenous reporter *ot907* [UNC-17::mKate]) and *ace-2* (fosmid-based reporter *otEx4432* [*ace-2::GFP*]) in multiple genetic backgrounds at L4 stage. *n* = 16–25 animals. For quantifications in **c**–**e**, box and whisker plots were used with the presentation of all data points. Unpaired *t*-test (two-sided) with Welch's correction was performed and *p*-values were annotated. Box boundaries indicate the 25th and 75th percentile. The limits indicate minima and maxima values, whereas center values (mean) are highlighted in red.
**f** Schematic summarizing our findings in cholinergic motor neurons that control locomotion. Source data are provided as a Source data file. Scale bar: 5 μm.

Due to its maintenance role in egg-laying MNs (Fig. 1g–i), we next asked whether *lin-39* is also required to maintain in the adult the cholinergic identity of MNs critical for locomotion. Again, we found that depletion of LIN-39 specifically during late larval and adult stages resulted in a decrease in the number of MNs expressing *unc-17/ChAT* (Fig. 2f, g), strongly suggesting a continuous LIN-39 requirement in these neurons.

### *lin-39* cooperates with *mab-5* and *unc-3* to control the identity of motor neurons

Compared to the strong effects observed in VC MNs (Fig. 1d), we observed modest effects on ACh pathway gene expression upon *lin-39* loss in MNs that control locomotion (Fig. 2c–f), suggesting additional factors partially compensate for *lin-39* in these neurons (Fig. 2h). To identify them, we followed a candidate approach. First, we reasoned that *mab-5* (*Antp, Hox6-8*), another Hox gene with continuous expression in a subset of *lin-39*-expressing MNs (Fig. 3a) (Supplementary Fig. 2)[26], may work together with *lin-39* to control NT identity features. To test this, we generated double *lin-39(n1760); mab-5(e1239)* mutant animals. All MNs that control locomotion are normally generated in this mutant[40]. However, we observed a decrease in the number of MNs expressing the cholinergic marker *unc-17/VAChT* in double *lin-39(n1760); mab-5(e1239)* mutants compared to single *lin-39(n1760)* mutant animals (Fig. 3d). Moreover, the residual expression of *unc-17/VAChT* in MNs of *lin-39; mab-5* double mutants suggests that, besides MAB-5, additional regulatory factors must cooperate with LIN-39. The transcription factor UNC-3, a member of the Collier/Olf/Ebf (COE) family, is selectively expressed in MNs that control locomotion (Fig. 3a) and is the only factor previously known to control their cholinergic identity[14]. Hence, if *unc-3* compensates for the combined loss of *lin-39* and *mab-5*, we would predict stronger effects on ACh pathway gene expression in MNs of triple mutant *unc-3(n3435); lin-39(n1760); mab-5(e1239)* animals than in single *unc-3* or double *lin-39; mab-5* mutants. Indeed, this was the case for *cho-1/ChT* and *ace-2/AChE* expression (Fig. 3b, c, e). Because animals carrying the strong loss-of-function *unc-3* allele *(n3435)*[41] display a striking reduction in the number of MNs expressing *unc-17/VAChT* (UNC-3 is absolutely required for *unc-17/VAChT*) (Fig. 3d), we used a hypomorphic (weaker loss-of-function) *unc-3* allele *(ot837)*[36] to test the notion of cooperation. Indeed, we observed stronger effects on *unc-17/VAChT* expression in *unc-3 (ot837); lin-39(n1760); mab-5(e1239)* compared to *unc-3 (ot837)*, and double *lin-39; mab-5* mutants (Fig. 3d). We extended the analysis of the hypomorphic *unc-3* allele *(ot837)* to *cho-1/ChT*, and again observed stronger effects in triple mutant animals (Fig. 3b, c, e). Although the magnitude of the observed effects differs in Hox and *unc-3* mutants, this genetic analysis suggests that *lin-39, mab-5*, and *unc-3* synergize to control the expression of several ACh pathway genes in MNs necessary for *C. elegans* locomotion (Fig. 3f).

### LIN-39 and UNC-3 act through distinct binding sites to activate ACh pathway gene expression

Interrogation of available ChIP-Seq datasets provided biochemical evidence of UNC-3 binding to the *cis*-regulatory regions of ACh pathway genes (*unc-17/VAChT, cho-1/ChT, ace-2/AChE*) (Fig. 4a, d) (Supplementary Fig. 3)[42]. These bound regions contain cognate sites for UNC-3, and their previous mutational analysis strongly suggested UNC-3 acts directly to activate ACh pathway gene expression in cholinergic MNs[14]. Next, we examined available ChIP-Seq datasets for LIN-39 and MAB-5[43], and witnessed coincident binding of LIN-39, MAB-5, and UNC-3 at *cis*-regulatory regions of *unc-17/VAChT, cho-1/ChT*, and *ace-2/AChE* (Fig. 4a, d, Supplementary Fig. 3). We hypothesized that these co-bound regions (putative enhancers) are sufficient to drive reporter (*yfp*) gene expression in MNs. Indeed, a 280bp-long fragment of the *cho-1 cis*-regulatory region, as well as two fragments (1000 and 125 bp) of the *unc-17* region, are sufficient to drive *yfp* expression in MNs

(Fig. 4a, b, d–f). The *yfp* expression driven by these putative enhancer regions (*cho-1_280bp, unc-17_1000bp, unc-17_125bp*) is *unc-3*- and Hox-dependent (Fig. 4b, c, e, f), as evidenced by a reduction in the number of MNs expressing the *yfp* reporter. Similar to our observations with an endogenous reporter for *unc-17* and a fosmid-based reporter for *cho-1* (Fig. 3c, d), we corroborated the notion of synergy between UNC-3 and Hox by using the hypomorphic *unc-3* allele *(ot837)*. Stronger effects were observed in *unc-3 (ot837); lin-39; mab-5* triple mutants compared to *unc-3(ot837)* single or *lin-39; mab-5* double mutant animals (Fig. 4c, e, f). Altogether, our enhancer analysis in *unc-3* and Hox mutants identified specific *cis*-regulatory regions that require both UNC-3 and Hox (LIN-39, MAB-5) input to drive *unc-17/VAChT* and *cho-1/ChT* expression in cholinergic MNs.

Within the *cho-1_280bp* and *unc-17_125bp* enhancer regions, a single UNC-3 binding site (COE motif) is necessary for reporter gene expression in MNs (Fig. 4a, d)[14]. We bioinformatically searched for the presence of consensus LIN-39-binding sites (GATTGATG), which, unlike MAB-5 sites, are well-defined in *C. elegans*[43]. We found a single LIN-39 binding site 60 bp upstream of the UNC-3 site in the *cho-1_280bp* region (Fig. 4a) and four LIN-39 sites flanking the UNC-3 binding site in the *unc-17_125bp* region (Fig. 4d). We tested the functional importance of the 4 LIN-39 sites by simultaneously deleting them in the context of transgenic reporter (*unc-17_125bp::yfp*) animals, and found a dramatic decrease in *yfp* expression in MNs (Fig. 4d). Altogether, this analysis combined with the ChIP-Seq results provide strong evidence that LIN-39 and UNC-3 bind directly to the *cis*-regulatory region of ACh pathway genes, and recognize distinct binding sites (Fig. 4g).

### A positive feed-forward loop (FFL) for the control of cholinergic motor neuron identity

Our findings suggest that Hox proteins LIN-39 and MAB-5, like UNC-3, directly control the expression of ACh pathway genes in MNs. However, LIN-39 and MAB-5 also bind extensively on the *cis*-regulatory region of *unc-3* (Fig. 5a), suggesting they directly regulate its expression as well. Indeed, expression of an endogenous *gfp* reporter for *unc-3* (UNC-3::GFP protein fusion) is significantly reduced in MNs of *lin-39* single mutants, and this effect is exacerbated in *lin-39; mab-5* double mutants (Fig. 5b, c). The number of MNs that express UNC-3::GFP is lower in *lin-39* and *lin-39; mab-5* animals (left graph in Fig. 5c). Compared to WT, we also observed a striking reduction in UNC-3::GFP fluorescence intensity in MNs of Hox mutant animals (34% reduction in *lin-39* and 43% in *lin-39; mab-5*) (middle graph in Fig. 5c), suggesting *lin-39* and *mab-5* are required for normal levels of UNC-3 expression.

The binding of LIN-39 and MAB-5 on the *unc-3* locus strongly suggests these Hox proteins control *unc-3* expression by acting at the level of transcription. Four lines of evidence support this possibility. First, CRISPR/Cas9-mediated mutation of a predicted LIN-39 binding site in the context of the endogenous *unc-3::gfp* reporter (UNC-3::GFP [LIN-39 site MUT]) resulted in decreased expression in MNs (right graph in Fig. 5a, c). Second, we quantified endogenous *unc-3* mRNA levels with single-cell resolution using RNA fluorescent in situ hybridization (RNA FISH)[44]. Compared to wild-type animals, we observed a consistent decrease of *unc-3* mRNA levels in cholinergic MNs of *lin-39; mab-5* mutants (Fig. 5d). Similar results were obtained with RT-PCR in animals carrying a hypomorphic *lin-39* allele (Supplementary Fig. 4). Third, a small 223 bp-long *cis*-regulatory element upstream of the first *unc-3* exon is sufficient to drive reporter gene (*mCherry*) expression in wild-type MNs (Fig. 5a, e). When we quantified the number of MNs expressing this reporter, as well as its fluorescence intensity, we observed statistically significant differences in Hox mutant animals (Fig. 5f, g). Like our observations with the UNC-3::GFP fusion (Fig. 5b, c), we observed a 28% reduction in fluorescence intensity of the *unc-3_223bp::mCherry* reporter in *lin-39* single mutants, which was

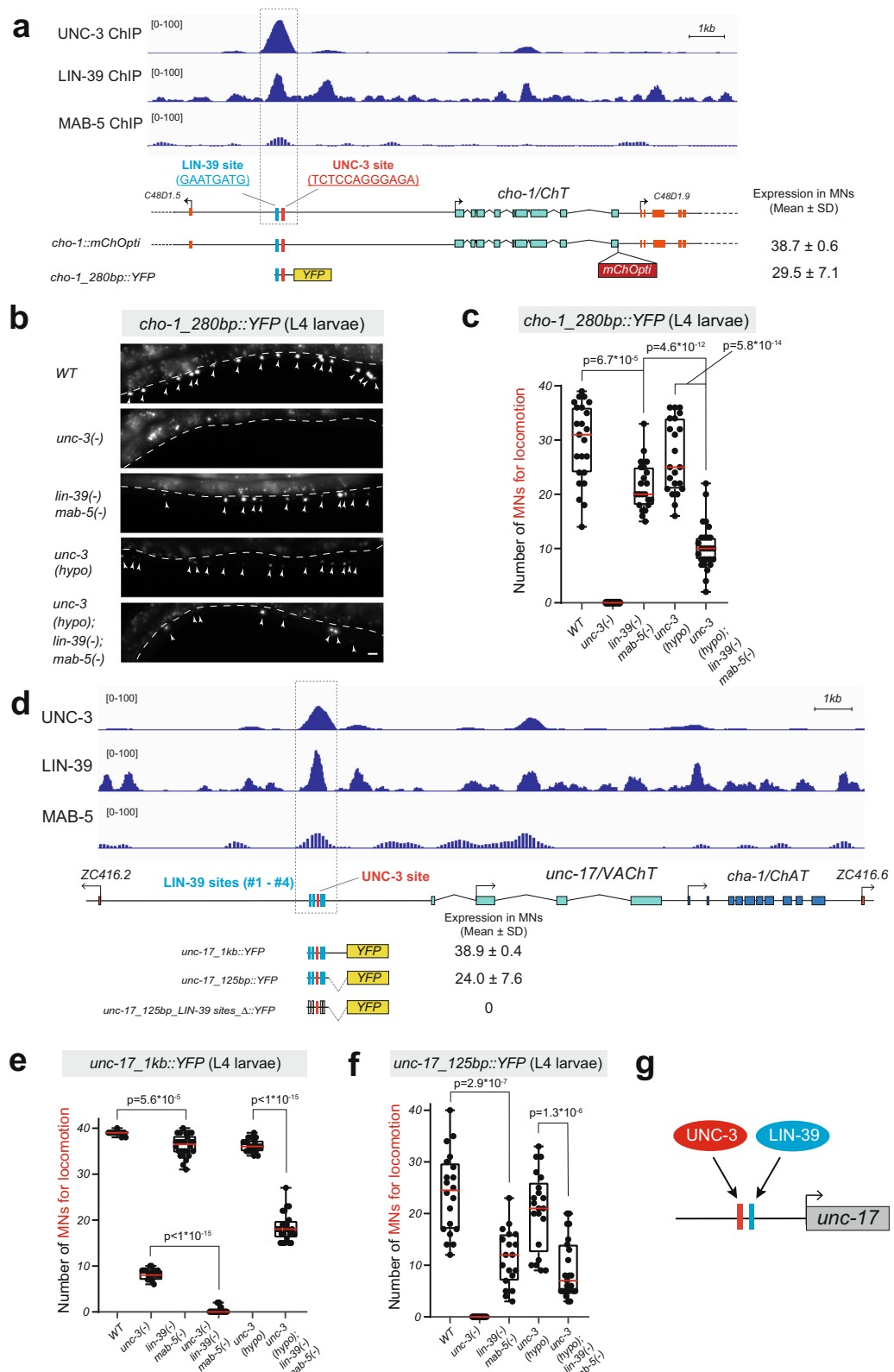

exacerbated (34% reduction) in *lin-39; mab-5* double mutants (Fig. 5g). Lastly, we generated two fosmid-based reporters, in which we replaced two different parts of the *unc-3* locus (exon 3–11 [ΔExon3–11], exon7–11 [ΔExon7–11]) with *mCherry* while leaving the upstream and downstream *cis*-regulatory elements intact (Fig. 5a). In both cases, we found a significant reduction in the number of *mCherry*-expressing MNs in *lin-39; mab-5* double mutant animals (Fig. 5h, i). We note that *unc-3*

expression is not completely abolished in motor neurons of *lin-39; mab-5* mutants (Fig. 5b–i), indicating that there must be additional, yet-to-be-identified factors that activate *unc-3* and compensate for the combined loss of *lin-39* and *mab-5*.

Altogether, Hox proteins (LIN-39, MAB-5) act directly to activate the expression of ACh pathway genes in MNs (Hox → ACh genes, Figs. 3 and 4). Further, Hox proteins act at the level of transcription to control

**Fig. 4 | LIN-39 and UNC-3 act directly to activate the expression of ACh pathway genes. a** Gene locus of *cho-1* with ChIP-seq tracks for UNC-3, LIN-39, and MAB-5. Molecular nature of *cho-1* reporter gene *cho-1::mChOpti* and *cho-1_280 bp::YFP* is shown. Total number of MNs expressing each reporter is indicated in the format of mean ± standard derivation (SD). Within a 280 bp *cis*-regulatory region (gray dashed line), an UNC-3 site and a predicted LIN-39 site are highlighted in red and blue, respectively. MAB-5 binding sites are not shown because they are not well-defined. **b** Representative images of *cho-1_280bp::YFP* expression at L4 in *WT, unc-3(n3435), lin-39(n1760) mab-5(e1239), unc-3(hypo),* and *unc-3(hypo); lin-39(n1760); mab-5(e1239)* animals. Arrowheads indicate MN nuclei because *cho-1_280bp::YFP* has a nuclear localization signal (NLS). YFP signal is shown in white for better contrast. Dashed white lines indicate the boundary of the intestine, which is auto-fluorescent in the green channel. **c** Quantification of the number of motor neurons expressing *cho-1 reporter* in panel (**b**). n = 18–27 animals. **d**. Gene locus of *unc-17* with ChIP-seq tracks for UNC-3, LIN-39, and MAB-5. Expression in motor neurons is quantified for *unc-17* reporter genes (*unc-17_1kb::YFP* and *unc-17_125bp::YFP),* as well as its mutated version with LIN-39 site deletions. Within a 125 bp *cis*-regulatory region framed by the gray dashed line, a UNC-3 site and 4 predicted LIN-39 sites are highlighted in red and blue, respectively. **e** and **f** Quantification of the number of motor neurons at L4 expressing the *unc-17_1kb::YFP* (**e**) and *unc-17_125bp::YFP* (**f**) reporters in *WT, unc-3(n3435), lin-39(n1760); mab-5(e1239), unc-3(n3435), lin-39(n1760); mab-5(e1239), unc-3(hypo),* and *unc-3(hypo); lin-39(n1760); mab-5(e1239)* animals. For all quantifications, box and whisker plots were used with the presentation of all data points. Box boundaries indicate the 25th and 75th percentile. The limits indicate minima and maxima values, whereas center values (mean) are highlighted in red. Unpaired *t*-test (two-sided) with Welch's correction was performed and p-values were annotated. n = 15–26 animals. **g** Schematic of *unc-17* regulation by UNC-3, LIN-39, and MAB-5. Red and blue rectangles indicate the binding sites of UNC-3 and LIN-39, respectively. Source data are provided as a Source data file. Scale bar: 5 μm.

*unc-3*, which also controls ACh genes (Hox → *unc-3* → ACh genes), thereby generating a positive feed-forward loop (FFL) (Fig. 5j).

### *lin-39* and *mab-5* maintain their expression through transcriptional autoregulation

How is Hox gene expression maintained in adult MNs? Available ChIP-Seq data show LIN-39 binding at its own locus in vivo (Fig. 6a), consistent with in vitro observations[45]. This raises the possibility of *lin-39* maintaining its own expression in MNs through transcriptional auto-regulation. To test this, we first determined which *cis*-regulatory elements of the *lin-39* locus are sufficient to drive reporter (*TagRFP*) expression in MNs. A 6.2 kb element upstream of the *lin-39* locus fused to *TagRFP* only shows low levels of expression in a small number of nerve cord MNs (~6) during larval stages (Fig. 6a). This prompted us to study intronic elements. When intron 1 of *lin-39* was fused to *TagRFP*, we detected robust *TagRFP* expression in ~23 nerve cord MNs, consistent with a previous study[45]. These *lin-39 intron 1::TagRFP* animals display continuous expression in both larval and adult MNs (Fig. 6a, b), similar to animals carrying the endogenous *lin-39* reporter allele (*lin-39::mNG::3xFLAG::AID*), enabling us to test the idea of transcriptional autoregulation. Indeed, the number of *TagRFP*-expressing MNs is significantly reduced in *lin-39* homozygous mutants carrying a null allele (*n1760*) (Fig. 6b), suggesting *lin-39* gene activity is necessary for its expression in MNs.

Because the *lin-39 (n1760)* allele removes gene activity starting in the early embryos, the above findings do not address whether LIN-39 is continuously required to maintain its expression at later stages. To test this, we again used the AID system and efficiently depleted LIN-39 protein levels during late larval stages (Fig. 6c). Upon auxin administration, we found a significant reduction in the number of MNs expressing the *lin-39 intron 1::TagRFP* reporter (Fig. 6d), suggesting *lin-39* is continuously required to maintain its own expression.

These findings pinpoint the entirety of intron 1 as a putative *cis*-regulatory region used by LIN-39 to directly control its own expression. Next, we generated transgenic reporter animals that split intron 1 into two fragments (832 and 776 bp). Animals carrying the 776 bp fragment (*lin-39 776bp::TagRFP*) revealed reporter gene expression in nerve cord MNs, but that was not the case for the 832 bp fragment (Fig. 6a). Within the 776 bp fragment, we found 11 predicted LIN-39 binding sites (motifs) (Fig. 6a), and employed CRISPR/Cas9 gene editing to simultaneously mutate all of them in the context of the endogenous *lin-39::mNG::3xFLAG::AID* reporter allele (SunyBiotech). This led not only to a decrease in the number of MNs expressing the endogenous *lin-39::mNG::3xFLAG::AID* sites 1-11 MUT reporter (Fig. 6e) but also to a reduction of its expression levels (Fig. 6f). These findings support a transcriptional autoregulation model where LIN-39 recognizes its cognate binding sites within intron 1 to regulate its expression in nerve cord MNs (Fig. 6i). Importantly, we observed functional consequences when *lin-39* autoregulation was selectively disrupted; the expression levels of

*cho-1/ChT* in cholinergic MNs were significantly reduced in animals carrying the *lin-39::mNG::3xFLAG::AID* sites 1–11 MUT allele (Fig. 6g).

To investigate which of the 11 LIN-39 sites are functionally important, we mutated two sites with the highest bioinformatic prediction scores (sites #1 and #9 in Fig. 6a, see the "Methods" section). Mutation of either site in the context of *lin-39 776bp::TagRFP* animals significantly decreased the number of *TagRFP*-expressing MNs (Fig. 6h), indicating that these sites are necessary for autoregulation.

Lastly, we asked whether the Hox gene *mab-5*, like *lin-39*, controls its own expression in MNs. Supporting this possibility, ChIP-Seq data show extensive MAB-5 binding at its own locus, and expression of a *mab-5::gfp* reporter is significantly reduced in MNs of *mab-5* mutant animals (Supplementary Fig. 5). Altogether, these findings uncover a positive feedback mechanism (transcriptional autoregulation) required to maintain Hox gene expression in MNs (Fig. 6j).

### UNC-3 (Collier/Ebf) prevents high levels of Hox gene (*lin-39, mab-5*) expression

Low levels of LIN-39 lead to a failure to maintain ACh pathway gene expression at appropriate levels (Figs. 1 and 2), whereas high levels of LIN-39 in cholinergic MNs result in locomotion defects[36]. Hence, we hypothesized that the levels of *lin-39* in cholinergic MNs must be tightly controlled to avoid detrimental effects on MN function (see the "Discussion" section). The findings described below suggest that tight control of Hox gene expression in MNs is achieved through a two-component mechanism, that is, Hox transcriptional autoregulation is counterbalanced by negative UNC-3 feedback.

Because UNC-3 ChIP binding peaks are observed in *lin-39* and *mab-5* loci (Fig. 7a, Supplementary Fig. 6a), we reasoned that UNC-3 acts directly to prevent high levels of Hox gene expression in cholinergic MNs. Multiple lines of evidence support this idea. First, the expression levels of the endogenous *lin-39::mNG::3xFLAG::AID* reporter are increased in MNs of *unc-3* mutant animals (Fig. 7b, c). Second, the *lin-39 intron 1::TagRFP* reporter showed increased *TagRFP* expression in MNs of *unc-3* null (Fig. 7d, e) and hypomorphic (Supplementary Fig. 7a) mutants. Lastly, RNA FISH revealed increased levels of *lin-39* mRNA molecules in individual MNs of *unc-3* mutants (Fig. 7f). Importantly, we extended this analysis to *mab-5*, and obtained similar results; *unc-3* gene activity is necessary to prevent high levels of *mab-5* expression in MNs (Supplementary Fig. 6b, c).

We next asked whether reducing UNC-3 levels by selectively disrupting the Hox input onto the *unc-3* locus can affect the expression of cholinergic identity genes. For this, we used the *UNC-3::GFP* LIN-39 site MUT allele, in which we observed lower levels of UNC-3 expression in motor neurons (Fig. 5a–c). We found that the expression levels of cholinergic identity genes (*unc-17/VAChT, cho-1/ChT*) are increased in motor neurons of animals carrying this allele (Supplementary Fig. 8), consistent with our model of low UNC-3 levels leading to a lack of efficient Hox gene repression (Fig. 7g).

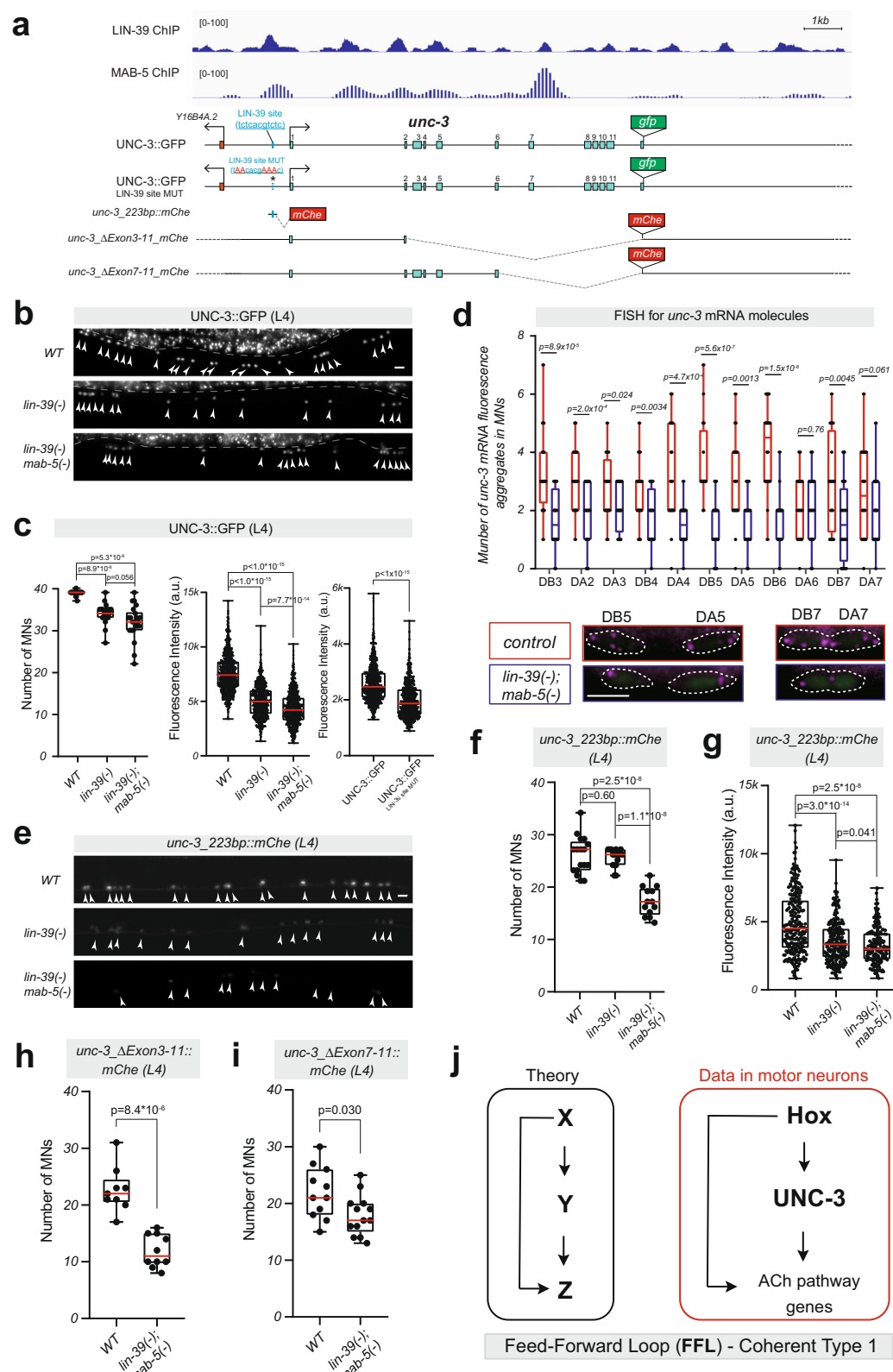

Taken together, our findings uncover an intricate gene regulatory network for the maintenance of cholinergic identity in *C. elegans* MNs. This network contains a positive FFL, where Hox proteins (LIN-39 and MAB-5) not only activate the expression of ACh pathway genes but also control the expression of *unc-3*, a critical regulator of cholinergic identity in MNs (Fig. 7g). Moreover, the top level of this FFL, i.e., Hox, is under homeostatic control, ensuring that optimal levels of Hox gene

expression are maintained in MNs. This is achieved through Hox transcriptional autoregulation counterbalanced by negative UNC-3 feedback (Fig. 7g).

## Discussion

Hox genes play fundamental roles in early patterning and body plan formation, but their functions in later stages of development and post-

**Fig. 5 | LIN-39 and MAB-5 control *unc-3* expression levels. a** ChIP-seq tracks for LIN-39 and MAB-5 on *unc-3* locus. Molecular nature of endogenous *unc-3* reporters (UNC-3::GFP and UNC-3::GFP [LIN-39 site MUT]) and transgenic *unc-3* reporters *(unc-3_223bp::mChe, unc-3 ΔExon3-11_mChe, unc-3 ΔExon7-11_mChe).* **b** Representative images of L4 animals carrying the endogenous *unc-3::gfp* reporter in *WT, lin-39(n1760),* and *lin-39(n1760); mab-5(e1239)* backgrounds. **c** Quantification of MN number (left graph, *n* = 19–22 animals) or fluorescence intensity (middle graph, *n* = 595 neurons) of UNC-3::GFP in WT, *lin-39(n1760), and lin-39(n1760); mab-5(e1239)* backgrounds. Right graph: Quantification of fluorescent GFP intensity in MNs of animals carrying the *UNC-3::*GFP and *UNC-3::GFP* [LIN-39 site MUT] endogenous reporters. *n* = 682 neurons. **d** Quantification of the number of *unc-3* mRNA fluorescence aggregates in individual cholinergic MNs. Single-molecule mRNA FISH for *unc-3* was performed in WT and *lin-39(n1760); mab-5(e1239)* animals at L1. For all quantifications, box and whisker plots were used; all data points shown. Unpaired *t*-test with Welch's correction was performed and *p*-values were annotated. *n* = 22 animals. Representative images of DB5, DA5, DB7, and DA7 neurons. WT (colored red) versus mutant (in blue) for quantification. Magenta: signal from Quasar 670 probe for *unc-3* mRNA, Green: *unc-17::gfp* reporter for cholinergic motor neurons. **e** Representative images of *unc-3_223bp::mChe* expression in L4 hermaphrodites in *WT, lin-39(n1760),* and *lin-39(n1760); mab-5(e1239).* **f** and **g** Quantification of MN numbers (**f**, *n* = 13–17 animals) or fluorescence intensity (**g**, *n* = 176 neurons) of *unc-3_223bp::mChe* in panel (**e**). For fluorescence intensity quantifications, single neurons expressing the mCherry reporters are plotted as individual data points. *n* > 600. In panel **g**, *n* > 200 neurons. **h** and **i** Quantification of the number of MNs expressing the *unc-3 ΔExon3-11_mChe* (**h**), and *unc-3 ΔExon7-11_mChe* (**i**) in WT and *lin-39(n1760); mab-5(e1239)* mutant animals. *n* = 10 animals. For all quantifications, box and whisker plots were used; all data points were presented. Box boundaries indicate the 25th and 75th percentile. The limits indicate minima and maxima values, whereas center values (mean) are highlighted in red. Unpaired *t*-test (two-sided) with Welch's correction was performed. **j** Schematic model of a coherent feed-forward loop (FFL) for the control of cholinergic identity. Source data are provided as a Source data file. Scale bars: 5 µm.

embryonic life remain largely unknown. In the context of the nervous system, Hox genes are known to control early events, such as specification, survival, and/or migration of progenitor cells and young post-mitotic neurons[18–22]. Here, we identify a critical role for Hox proteins during the later stages of nervous system development and post-embryonic life, critically extending their functional repertoire beyond early patterning. Using *C. elegans* nerve cord MNs as a model, we found that Hox gene activity is continuously required from development through adulthood for the control of NT identity, a core element of neuronal function. Moreover, Hox genes operate in a positive feed-forward loop to safeguard the cholinergic identity of MNs: they not only control directly ACh pathway genes (e.g., *unc-17/VAChT, cho-1/ChT*), but also the expression of *unc-3* (*Collier/Ebf*), a master regulator of MN identity[14]. Lastly, we propose a homeostatic mechanism that maintains Hox gene expression in MNs at optimal levels, thereby ensuring the robust expression of cholinergic identity determinants throughout life.

## Hox genes *lin-39* (*Scr/Dfd, Hox4-5*) and *mab-5 (Antp, Hox6-8)* act as terminal selectors of cholinergic MN identity

Transcription factors that are continuously expressed in individual neuron types and act directly to co-regulate the expression of NT pathway genes have been termed "terminal selectors"[46–48]. Terminal selectors have been described to date for various neuron types in *C. elegans*, flies, simple chordates, and mice, indicating the evolutionary conservation of terminal selector-based mechanisms for the control of NT identity[49,50]. The sustained expression of terminal selectors in specific neuron types suggests they are continuously required to maintain NT identity. However, a continuous requirement has been experimentally demonstrated only for a handful of terminal selectors to date[51].

Recent work in *C. elegans*, flies, and mice indicates that certain Hox genes are continuously expressed, from development through adulthood, in distinct neuron types[23–29]. Functional studies in peptidergic neurons in *Drosophila*, as well as hindbrain and spinal neurons in mice, showed that Hox proteins control early facets of neuronal development (e.g., specification, survival, migration, connectivity)[18–22,52], but whether they function as terminal selectors remains unknown—in part due to a lack of temporally controlled studies for Hox gene inactivation later in life. Through constitutive (genetic null alleles) and post-embryonic (inducible protein depletion) approaches, we found that the Hox gene *lin-39* is continuously required to maintain the cholinergic identity of *C. elegans* MNs. Biochemical (ChIP-Seq) and genetic evidence strongly suggest that LIN-39 and another Hox protein (MAB-5) act directly to co-regulate the expression of genes that encode integral components of the ACh biosynthetic pathway (*unc-17/VAChT, cho-1/ChT, ace-2/AChE*). Collectively, these findings reveal that Hox proteins can act as terminal selectors, uncovering a noncanonical role for these highly conserved transcription factors.

Because Hox genes are expressed in various neuron types in *C. elegans*[29], we surmise they may function as terminal selectors in other neurons as well. Supporting this notion, expression of a single marker of serotonergic identity (*tph-1*, tryptophan hydroxylase 1 [TPH1]) is affected in CP neurons of *lin-39* mutant animals[53,54]. Similarly, expression of a dopaminergic identity marker (*cat-2*/tyrosine hydroxylase [TH]) is affected in tail sensory neurons of animals lacking activity of the posterior Hox gene *egl-5* (*AbdB*)[55]. Future work is needed, however, to establish whether in those neurons LIN-39 and EGL-5 act as bona fide terminal selectors.

## Hox and other subfamilies of homeodomain proteins act as terminal selectors

Homeodomain proteins are defined by a 60 amino acid motif (homeodomain) that directly contacts DNA[56]. Based on sequence similarities and/or the presence of other domains, several subfamilies of homeodomain proteins (e.g., HOX, LIM, POU, PRD) have been identified in every animal genome. In *C. elegans*, accumulating evidence suggests that members of LIM, POU, and PRD subfamilies can function as terminal selectors in specific neuron types[35,57]. Here, we show that members of the HOX subfamily can act as terminal selectors in cholinergic MNs. A synthesis of our findings and previous studies on LIM, POU, and PRD proteins reveals an overarching theme: a single transcription factor family, the homeodomain proteins, is broadly used in the *C. elegans* nervous system to control neuronal identity by acting as terminal selectors. Supporting the evolutionary conservation of this theme, a number of LIM and POU homeodomain transcription factors can function as terminal selectors in the mouse nervous system[9,10,58–60].

## Hox proteins cooperate with the terminal selector UNC-3 (Collier/Ebf) to control motor neuron cholinergic identity

Besides NT identity, terminal selectors are known to control additional neuron type-specific features of terminal differentiation, such as the expression of ion channels, neuropeptides, NT receptors, and cell adhesion molecules[46,48]. We find this to be the case for LIN-39. In sex-specific MNs that control egg laying, LIN-39 is required for continuous expression of ACh pathway genes and additional terminal differentiation markers (e.g., *glr-5/GluR, ida-1/PTPRN*). Similarly, in MNs necessary for locomotion, LIN-39 controls ACh pathway genes (this study) and a handful of terminal differentiation markers (e.g., *del-1/* sodium channel SCNN1, *slo-2/*potassium−sodium-activated channel KCNT)[26,36]. But how can the same Hox protein (LIN-39) operate as a terminal selector in two different types of MNs?

Our findings support the idea of LIN-39 cooperating with distinct transcription factors in different MN types. In MNs that control locomotion, LIN-39 synergizes with MAB-5 and UNC-3; the latter is known to function as a terminal selector in these neurons[14]. Mechanistically,

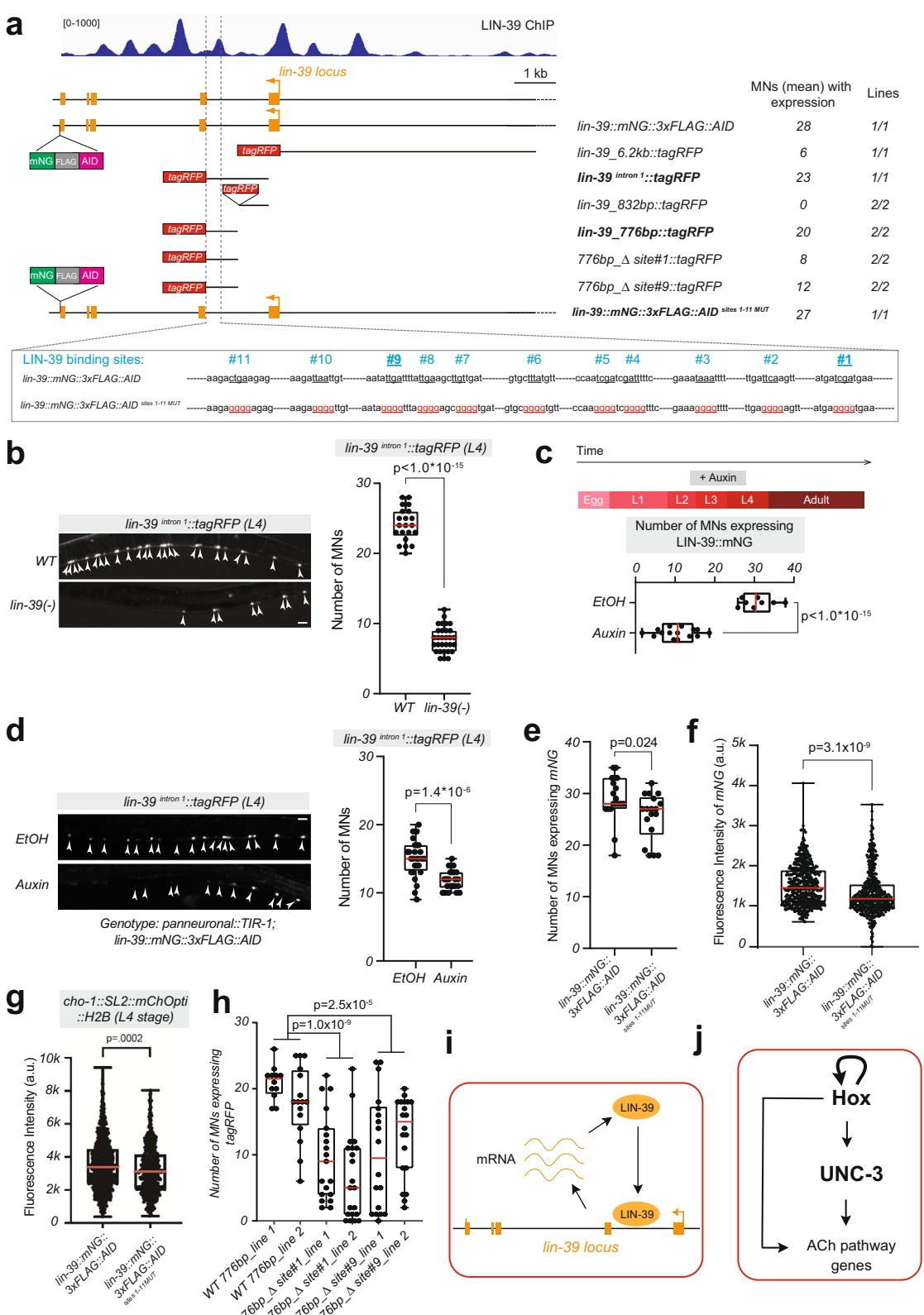

LIN-39 and UNC-3 control the same ACh pathway genes by recognizing distinct binding sites. However, whether LIN-39 and UNC-3 are being recruited to these sites in an additive or cooperative manner remains unresolved. In MNs that control egg-laying[61], UNC-3 is not present[14]. HLH-3 (bHLH protein of Achaete-Scute family) and LIN-11 (LIM homeodomain protein) are expressed in VC neurons[62,63], constituting putative LIN-39 collaborators for the control of cholinergic identity in VC.

## Hox and UNC-3 in a positive feed-forward loop (FFL): a form of redundancy engineering to ensure robust expression of cholinergic identity genes

Complex gene regulatory networks are composed of simple gene circuits called "network motifs"[64]. One of the most widely used motifs is the feed-forward loop (FFL), where transcription factor X activates a second transcription factor Y, and both activate their shared target

**Fig. 6 | Transcriptional autoregulation of *lin-39* in motor neurons. a** ChIP-seq tracks for LIN-39 on *lin-39* locus. Molecular nature of endogenous *lin-39* reporters (*lin-39::mNG::3xFLAG::AID* and *lin-39::mNG::3xFLAG::AID^sites1-11MUT*) and transgenic *lin-39* reporters (*lin-39_6.2kb::tagRFP, lin-39^intron1::tagRFP, lin-39_832bp::tagRFP, lin-39_776bp::tagRFP, 776bp_Δsite#1::tagRFP, 776bp_Δsite#9::tagRFP*). The mean value of MNs that express each reporter is indicated on the right. The precise location of the 11 predicted LIN-39 binding sites is shown, and their mutated version. **b** Representative images and quantification of the number of MNs expressing *lin-39^intron1::tagRFP* in *WT* and *lin-39(n1760)* mutants at L4. *n* = 21–30 animals. **c** Depletion of LIN-39::mNG::3xFLAG::AID upon auxin. Worms expressing the *lin-39::mNG::3xFLAG::AID* together with TIR protein in all neurons (driven by pan-neuronal promoter *otti28*) were grown on normal OP50 plates until L2 stage and then transferred to plates containing either auxin or EtOH (as control). Animals were imaged at L4. Quantification of LIN-39::mNG::3xFLAG::AID expression in motor neurons upon Auxin or EtOH treatment. *n* = 8–15 animals. **d** Representative images and quantification of the number of MNs expressing *lin-39^intron1::tagRFP* treated with EtOH or auxin. *n* = 26-28 animals. **e** and **f** Quantification (L4 stage) of the number of MNs expressing *mNG* (**e**, *n* = 18–19 animals) or the fluorescence intensity (**f**, *n* = 459 neurons) of *mNG* in animals carrying the *lin-39::mNG::3xFLAG::AID* and *lin-39::mNG::3xFLAG::AID^sites1-11MUT* alleles. In panel **f**, single neurons expressing *mNG* are plotted as individual data points. *n* > 400 cells. a.u. arbitrary units. **g** Quantification of fluorescent intensity of *cho-1::SL2::mChOpti::H2B* expression in MNs of *lin-39::mNG::3xFLAG::AID* or *lin-39::mNG::3xFLAG::AID^sites1-11MUT* animals (L4 stage). **h** Quantification of the number of MNs expressing the *lin-39_776bp::tagRFP* reporter (intact and mutated version with LIN-39 site#1 or #9 mutated). Two independent transgenic lines for each version of the reporter were used. *n* = 12–21 animals. For all quantifications, box and whisker plots were used; all data points were presented. Box boundaries indicate the 25th and 75th percentile. The limits indicate minima and maxima values, whereas center values (mean) are highlighted in red. Unpaired *t*-test (two-sided) with Welch's correction was performed. *n* > 15 animals. **i** Schematic model of *lin-39* transcriptional autoregulation. **j** Schematic model showing the autoregulation of *lin-39* embedded at the top of a coherent FFL. Source data are provided as a Source data file. Scale bars: 5 µm.

gene Z (Fig. 5j). Our findings uncovered a FFL in cholinergic MNs, where a Hox protein (LIN-39) activates UNC-3, and both activate their shared targets (ACh pathway genes) (Fig. 5j). FFLs have been described in transcriptional networks across species[64–66]. Based on systems biology classifications, there are 8 different types of FFLs−each characterized by the signs ("+" for activation, "−" for repression) of the transcriptional interactions within the motif. The FFL in MNs is coherent type 1 because all interactions are activating (Fig. 5j).

Why is there a need for this type of FFL in *C. elegans* MNs? Computational models and studies in bacteria indicate that a coherent type 1 FFL can ensure robust gene expression against perturbations through a process called "sign-sensitive delay"[66–68]. That is, a perturbation can decrease the levels of transcription factor X (LIN-39), but Z (ACh pathway genes) responds only at a delay once X levels decrease. The delay is due to the presence of Y (UNC-3). After X is decreased, it takes time for Y levels to decrease (depending on the degradation rate of Y) to a level insufficient to activate Z. Hence, the coherent type 1 FFL can be viewed as a form of redundancy engineering (or filter) to protect the shared target genes (Z) from fluctuations in the input (transcription factor X). Based on computational modeling and functional studies in bacteria[66–68], we propose that, in our system, LIN-39 (X) and UNC-3 (Y) operate in a coherent type 1 FFL to maintain robust expression levels of ACh pathway genes in MNs throughout life (Fig. 5j). Lastly, we note that coherent and incoherent FFLs can serve various functions in the developing nervous system. For example, they can act as developmental timers (by allowing a given transcription factor to control different genes at successive developmental stages)[69–71], or as means to generate neuronal subtype diversity[72,73].

The FFL we identified in *C. elegans* MNs forms the backbone of a transcriptional network, but it does not function in isolation. Rather, it is embedded within at least two additional loops: (a) a positive feedback loop that activates LIN-39 expression through transcriptional autoregulation, and (b) negative feedback provided by UNC-3 to reduce LIN-39 expression (Fig. 7g). The significance of both is discussed below.

Consistent with our observations on Hox (LIN-39, MAB-5) and *unc-3*, two other *C. elegans* Hox proteins (CEH-13, EGL-5) control the transcription of another terminal selector (*mec-3*) in peripheral touch receptor neurons[74]. Although a FFL was not identified in that cellular context, the findings on *mec-3* and *unc-3* (this study) strongly suggest that Hox-mediated transcriptional control of terminal selectors may be a broadly applicable strategy to ensure the robustness of gene expression in post-mitotic neurons.

### A two-component design principle for homeostatic control of Hox gene expression in motor neurons

Mechanisms for the initiation of Hox gene expression are well-studied. It is known, for example, that during embryonic patterning, Polycomb Group (PcG) and trithorax Group (trxG) genes determine the levels and spatial expression of Hox genes[75–79]. It is also known that, during early nervous system development, morphogenetic gradients initially establish Hox gene expression, and this is further refined via Hox cross-regulatory interactions[22,80]. However, the mechanisms that maintain Hox gene expression in post-mitotic neurons during later developmental and post-embryonic stages remain unknown.

Our constitutive (genetic null alleles) and temporally controlled (AID system) approaches combined with mutagenesis of Hox binding sites within the endogenous *lin-39* locus strongly suggest that LIN-39 maintains its own expression in MNs through transcriptional autoregulation (Fig. 6i). Subsequently, LIN-39 is required to maintain the expression of ACh pathway genes by operating at the top of a coherent FFL (Fig. 6j). However, the *lin-39* expression levels are critical: low levels lead to a failure to maintain ACh pathway gene expression (Figs. 1 and 2), whereas high levels of LIN-39 in cholinergic MNs cause locomotion defects and "mixed neuronal identity", i.e., genes normally expressed in GABA neurons became ectopically expressed in cholinergic MNs[36]. We found that *lin-39* expression in MNs is under homeostatic control. The terminal selector UNC-3 prevents high levels of *lin-39* expression, thereby counterbalancing the positive effect of the transcriptional autoregulation of *lin-39*. From the perspective of Hox binding site affinity[81,82], this mechanism may enable LIN-39 (when present at optimal levels) to bind at high-affinity sites in the *cis*-regulatory region of ACh pathway genes. The negative UNC-3 feedback prevents high levels of LIN-39, potentially avoiding LIN-39 binding to low-affinity sites in the *cis*-regulatory region of alternative (e.g., GABA) identity genes.

Altogether, we identified a two-component design principle (positive feedback of LIN-39 to itself, negative feedback from the terminal selector UNC-3) for the homeostatic control of Hox gene expression in motor neurons. These two components are embedded in a coherent FFL, which we propose to be necessary for the maintenance of NT identity (Fig. 7g). Because Hox genes are continuously expressed in the adult fly, mouse, and human nervous systems, the gene regulatory architecture described here for the control of NT identity may constitute a broadly applicable principle.

## Methods

### *C. elegans* strains

Worms were grown at 15, 20, or 25 °C on nematode growth media (NGM) plates seeded with bacteria (*E. coli* OP50) as food source[83]. Mutant alleles used in this study: *unc-3 (n3435) X, lin-39 (n1760) III, mab-5 (e1239) III, ced-3 (n1286) IV.* CRISPR-generated alleles: *unc-3 (ot837 [unc-3::mNG::3xFLAG::AID]) X, lin-39 (kas9 [lin-39::mNG::3xFLAG::AID]) III.*

All reporter strains used in this study are shown in Supplementary Data 1.

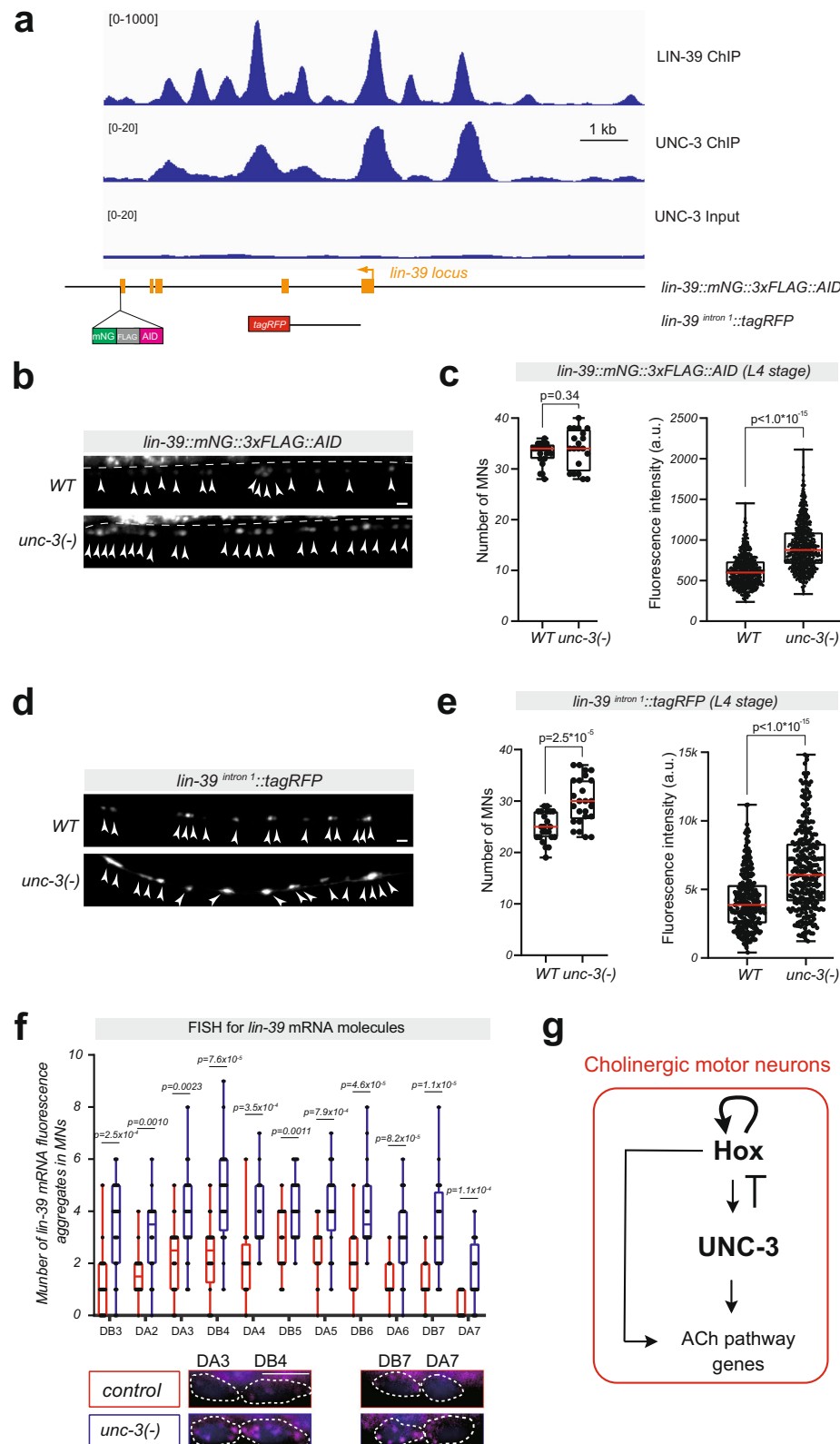

## Generation of transgenic reporter animals

Reporter gene fusions for *cis*-regulatory analysis were made using either PCR fusion[84] or Gibson Assembly Cloning Kit (NEB #5510S). Targeted DNA fragments were fused (ligated) to the *tagrfp*-coding sequence, which was followed by *unc-54 3' UTR*. Mutations or deletions of LIN-39-binding sites were introduced via PCR muta-genesis. The product DNA fragments were either injected into young adult *pha-1(e2123)* hermaphrodites at 50 ng/µl using *pha-1* (pBX plasmid) as co-injection marker (50 ng/µl) and further selected for survival, or injected into young adult N2 hermaphrodites at 50 ng/µl (plus 50 ng/µl pBX plasmid) using *myo-2::gfp* as co-injection marker (3 ng/µl) and further selected for GFP signal.

**Fig. 7 | UNC-3 (Collier/Ebf) prevents high levels of *lin-39* expression. a** Gene locus of *lin-39* with ChIP-seq tracks for UNC-3 and LIN-39. Molecular nature of *lin-39* reporter gene *lin-39::mNG::3xFLAG::AID* and *lin-39^intron1^::tagRFP* is shown. **b** and **c** Representative images (**b**) and quantifications (**c**) of the number of MNs (*n* = 20 animals) expressing *lin-39::mNG::3xFLAG::AID* and the fluorescence intensity (*n* = 660 neurons) in *WT* and *unc-3(n3435)* animals at L4. **d** and **e** Representative images (**d**) and quantifications (**e**) of the number (*n* = 23 animals) and the fluorescence intensity (*n* = 264 neurons) of MNs expressing *lin-39^intron1^::tagRFP* in *WT* versus *unc-3(n3435)* animals at L4. In panels **c** and **e**, single neurons expressing the indicated reporter gene are evaluated for fluorescence intensity and plotted as individual data points in the graph. *n* > 600 neurons. a.u. arbitrary unit. **f** Quantification of the number of *lin-39* mRNA fluorescence aggregates in individual cholinergic MNs. Single-molecule mRNA FISH for *lin-39* (using a Quasar 670

fluorescent probe) was performed in WT and *unc-3(n3435)* animals at L1 (*n* = 20 animals). For all quantifications, box and whisker plots were used with the presentation of all data points. Box boundaries indicate the 25th and 75th percentile. The limits indicate minima and maxima values, whereas center values (mean) are highlighted in red. Unpaired *t*-test (two-sided) with Welch's correction was performed and *p*-values were annotated. *n* > 20 animals. Representative images of DA3, DB4, DB7, and DA7 are shown. WT data is colored red versus mutant in blue for quantification. Images are shown in merged channels. Magenta: signal from Quasar 670 probe for *lin-39* mRNA, Blue: DAPI for nuclei staining. **g** Schematic model of the gene regulatory network for maintenance of cholinergic identity in *C. elegans* motor neurons. Source data are provided as a Source data file. Scale bars: 5 μm.

## Targeted genome engineering

CRISPR/Cas9 genome editing was employed to generate: (a) PHX2477 strain carrying the *lin-39(syb2477 [lin-39_site1-11_MUT::mNG::3xFLAG::AID])* III allele, and (b) PHX5816 strain carrying the *unc-3(syb5816 [unc-3_LIN-39 site_MUT_GFP])* X allele. Gene editing was performed by SunyBiotech.

## Temporally controlled protein degradation

AID-tagged proteins are conditionally degraded when exposed to auxin in the presence of TIR1[37]. Animals carrying auxin-inducible alleles of *lin-39 [kas9 [lin-39::mNG::AID]]* were crossed with *ieSi57* animals that express TIR1 ubiquitously or with *otTi28* animals that express TIR-1 pan-neuronally. Auxin (indole-3-acetic acid [IAA], Catalog number A10556, Alfa Aesar) was dissolved in ethanol (EtOH) to prepare 400 mM stock solutions which were stored at 4 °C for up to one month. NGM agar plates were poured with auxin or ethanol added to a final concentration of 4 mM and allowed to dry overnight at room temperature. Plates were seeded with OP50 bacteria. To induce protein degradation, worms of the experimental strains were transferred onto auxin-coated plates and kept at 25 °C. As a control, worms were transferred onto EtOH-coated plates instead. Auxin solutions, auxin-coated plates, and experimental plates were shielded from light.

## Single molecule RNA in situ hybridization (sm RNA-FISH)

Egg preparation was performed before harvesting the synchronized L1 worms. Next, worms were fixed with 4% paraformaldehyde (PFA), washed twice with PBS, and permeabilized with 70% ethanol at 4 °C overnight. Then, samples were hybridized with diluted mRNA probes (Stellaris; *unc-3*: unc-3 best Quasar 670 (SMF-1065-5); *lin-39*: lin-39 best Quasar 670 (SMF-1065-5)) in hybridization buffer (Stellaris#174; RNA FISH hybridization buffer (SMF-HB1-10)) following manufacturing protocols. Finally, samples were stained with DAPI, washed, and equilibrated with GLOX buffer. Worms were imaged at a fluorescence microscope (Zeiss, Axio Imager.Z2).

## Real-time PCR assay for *unc-3* and *lin-39* expression level analysis

Synchronized L4 stage *C. elegans* animals were collected from different genotypes (see figure legends), and mRNA was extracted. cDNA library was prepared using the Superscript first strand cDNA synthesis kit (Invitrogen #11904-018). RT-PCR TaqMan assays for the genes *unc-3* (assay ID: Ce02402729_g1), *lin-39* (assay ID: Ce02406896_m1), and *pmp-3* (Ce02485188_m1) were performed, and the expression level of *unc-3* and *lin-39* were determined in each genotype after normalizing to the expression of the housekeeping gene *pmp-3*.

## Microscopy

Worms were anesthetized using 100 mM of sodium azide (NaN₃) and mounted on a 4% agarose pad on glass slides. Images were taken using an automated fluorescence microscope (Zeiss, Axio Imager.Z2). Acquisition of several Z-stack images (each ~1 μm thick) was taken with

Zeiss Axiocam 503 mono using the ZEN software (Version 2.3.69.1000, Blue edition). Representative images are shown following max-projection of 1–8 μm Z-stacks using the maximum intensity projection type. Image reconstruction was performed using Image J software[85].

## Motor neuron identification

Motor neurons were identified based on a combination of the following factors: (i) co-localization with fluorescent markers with known expression pattern, (ii) invariant cell body position along the ventral nerve cord, or relative to other MN subtypes, (iii) MN birth order, and (iv) number of MNs that belong to each subtype.

## Bioinformatic analysis

To predict the UNC-3-binding site (COE motif) in the *cis*-regulatory region of target genes, we used the MatInspector program from Genomatix[86]. The position weight matrix (PWM) for the LIN-39-binding site is cataloged in the CIS-BP (Catalog of Inferred Sequence Binding Preferences database)[87]. To identify putative LIN-39 sites, we used FIMO (Find Individual Motif Occurrences)[88], which is one of the motif-based sequence analysis tools of multiple expectation maximization for motif elicitation (MEME) bioinformatics suites (http://meme-suite.org/). The *p*-value threshold for the analysis of LIN-39 target genes was set at *p* < 0.001 for *unc-17* and *cho-1*, and at *p* < 0.01 for *lin-39*.

## Fluorescence intensity (FI) quantification

To quantify FI of individual MNs in the VNC, images of worms from different genetic backgrounds were taken with identical parameters through full-thickness Z-stacks that cover the entire cell body. Image stacks were then processed and quantified for FI via FIJI. The focal plane in Z-stacks that has the brightest FI was selected for quantification to minimize background signals. Cell outline was manually selected, and FIJI was used to quantify the FI and area to get the mean value for FI. After quantifying the FI of all MNs of interest, representative average image background FI was quantified and subtracted from the individual MN FI quantifications to get the mean FI in arbitrary units (a.u.).

## Statistical analysis and reproducibility

For quantification, box and whisker plots were adopted to represent the quartiles in graph. The box includes data points from the first to the third quartile value with the horizontal line in box representing the mean value. Upper and lower limits indicate the max and min, respectively. Unpaired *t*-test with Welch's correction was performed and *p*-values were annotated. Visualization of data and *p*-value calculation were performed via GraphPad Prism Version 9.2.0 (283). Each experiment was repeated twice.

## Reporting summary

Further information on research design is available in the Nature Research Reporting Summary linked to this article.

## Data availability

The data supporting the findings of this study are included in the figures and supporting files. The ChIP-Seq data for UNC-3, LIN-39 and MAB-5 used in this study are available in the NCBI Gene Expression Omnibus (GEO) database under accession codes: GSE143165 GSE25785 GSE15625. Source data are provided with this paper.

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

## Acknowledgements

We thank the Caenorhabditis Genetics Center (CGC), which is funded by NIH Office of Research Infrastructure Programs (P40 OD010440), for providing strains. We are grateful to Oliver Hobert and members of the Kratsios lab (Edgar Correa, Nidhi Sharma, Filipe Marques, Manasa Prahlad, Anthony Osuma) for their comments on this manuscript. This work was funded by two NIH grants (R01 NS116365-01, R01 NS118078-01) to P.K.

## Author contributions

W.F. Conceptualization, data curation, investigation, visualization, methodology, writing—review and editing; H.D., J.J.S. Formal analysis, validation, investigation, P.K. Conceptualization, supervision, investigation, funding acquisition, project administration, writing—original draft, review, and editing.

## Competing interests

The authors declare no competing interests.
