## [Peer Review File · Nature Communications]

Maintenance of neurotransmitter identity by Hox proteins through a homeostatic mechanismREVIEWER COMMENTS

Reviewer #1 (Remarks to the Author):

This study investigates the functional relevance of sustained expression of Hox transcription factors in adult neurons in the ventral cord of *C. elegans*. It was previously observed that neuronal Hox protein expression persists in to adulthood, but its functional significance has not been thoroughly investigated. This paper demonstrates that this extended expression of the Hox transcription factor LIN-39 is necessary for expression of multiple indicators of cholinergic fate in ventral cord neurons that control egg laying and locomotion in *C. elegans* hermaphrodites. The authors additionally reveal a role for the Hox protein MAB-5 and the OE/Collier transcription factor UNC-3 in maintaining cholinergic identity in these neurons. The authors present evidence for direct regulation of cholinergic genes by all three transcription factors, as well as providing evidence for a co-regulatory feedforward loop among the Hox genes and UNC-3.

This work builds on previous studies from the authors' lab demonstrating that UNC-3 interacts with LIN-39 and MAB-5 in its role as a terminal selector for motor neuron identity in *C. elegans*, as well as on a body of previous work that has explored the role of LIN-39 and MAB-5 in specifying male-specific neurons, vulval precursors, and other cell fates in *C. elegans*. A major strength of this paper is that it makes use of reagents that allow temporal control of LIN-39 expression, thus bypassing the previous hurdles presented by the important role of LIN-39 in regulating cell division and survival earlier in larval development. It also makes good use of single-copy endogenous reporters when possible, as well as publicly available ChIP data to make a strong case for the findings presented here.

The work here is solid, significant, and thorough, as it provides mechanistic detail to models hypothesized based on genetic data and expression of multicopy reporter transgenes. I would like to have seen the authors take more care in acknowledging previous work done on LIN-39 and MAB-5, particularly their post-embryonic roles, and have outlined some examples below:

1) In the introduction (line 111), the statement "whether and how Hox proteins control neurotransmitter identity is unknown," omits previous work published by Clark et al., 1993 and Salser et al., 1993, demonstrating the requirement for LIN-39 in specifying serotonergic fate of CP neurons in males. I think omitting the "whether" from this statement would resolve this issue.

2) Also in the male CPs, the work of Hunter and Kenyon (1995) demonstrated that promotion of serotonergic fate can occur as late as the L4 larval stage, suggesting a late/possible ongoing role for LIN-39 in this process as well.

3) In the results, the authors mention that Hox proteins can act as developmental timers. Salser and Kenyon (1996) found that MAB-5 seems to play such a role in the male seam lineage, acting at various stages throughout larval development to control timing of neuronal specification.

The methodology employed is sound and meets the expectations for the field. The authors' combined use of reporters, ChIP, genetics and FISH make a strong case for their findings. They are overall careful in drawing conclusions, and the conclusions follow from the data presented. I have a few specific suggestions about the results section.

1) Line 144 of results: The authors should acknowledge that fosmid reporters are multicopy and do tend to represent overexpression.

2) In the section "LIN-39 and UNC-3 act through distinct binding..." , I found it somewhat confusing that that text states that the regulatory regions of cholinergic genes show (by previously published ChIP) binding to LIN-39, MAB-5, and UNC-3, but that the sites labeled in Figure 4 are only for LIN-39

and UNC-3. It would be helpful to provide clarification in the figure legend for the focus on LIN-39 sites, which I believe was due to the site being well-defined, as stated in the text.

3)Line 291 of results: please change to "A 6.2kb element upstream of the lin-39 locus," for clarity.

4)Results section "Hox genes lin-39 and mab-5 maintain their expression...". It has been previously shown by Wagmeister et al. (2006) that expression of lin-39 reporters in the ventral cord is regulated by an intronic element. They further showed (via EMSA) that LIN-39 binds to hox consensus sites in this region. Although this paper's finding demonstrates in vivo binding via ChIP, the previous work should be acknowledged.

5)Figure 2: the affect in F in adults upon auxin treatment is subtle. It would be helpful to see a representative image for F.

6)For all figures showing box and whisker plots for motor neuron number, add means to graphs or figure legends (the dots obscure the mean line in many of these).

Reviewer #2 (Remarks to the Author):

In this work Feng et al. expand on previous work from the group (mainly Feng eLife 2020) to further characterize the gene regulatory networks that establish and maintain motorneuron identity in *C. elegans*.

Previous work from the authors showed that hox LIN-39 and MAB-5 are required to induce and maintain expression of cholinergic Motorneuron (MN) terminal identity genes, this observation is now expanded to include Acetylcholine biosynthesis pathway gene regulation which is described both in locomotor MN and vulva muscle MN.

Authors also show that MAB-5 and LIN-39 act directly on the activation of these target genes binding to similar genomic regions as UNC-3 terminal selector, suggesting these factors act together to control MN fate. They also describe that HOX genes activate unc-3 expression, creating a feed forward loop. In addition HOX also autoregulate their expression. As discussed by the authors, these regulatory motifs are commonly found in cell fate identity networks.

Finally, and most interestingly they also find a repressive action of UNC-3 on lin-39 and mab-5 expression. This described network tightly regulates HOX gene expression levels and this regulation is key for proper MN effector gene expression, as low levels of hox leads to MN gene expression defects but, as authors showed in previous work, high levels of LIN-39 lead to ectopic expression of effector genes from other motorneuron classes. This tight level control is defined by the authors as "homeostatic control".

The manuscript is well-written and well-organized, it is easy to follow, experiments are solid and questions are addressed combining several strategies what makes the work very robust.

The description of the role of HOX genes in neuron fate maintenance is interesting as well as the importance of tight control of HOX gene expression levels.

My main concern is related to the use of "homeostatic" control to define this mechanism, which I think deserves further work to justify its use in contrast to other terms such as robust gene expression mechanism.

As I understand, homeostatic systems respond to a perturbation with compensatory feedback such that the set point activity of the system is precisely reestablished.

Current data in the paper is very attractive but in my opinion is still not enough to fully support the use of the term homeostatic control.

Indeed the expectation for homeostatic control would be that ectopic lin-39 expression (performed in Feng 2020 eLife) would lead to increase UNC-3 levels that will in turn decrease expression of lin-39 endogenous locus. This could be experimentally tested using Q-PCR for lin-39 3'UTR to distinguish

endogenous gene from lin-39 cDNA overexpression.

Alternatively, unc-3 hypomorphic alleles might lead to an increase of lin-39 expression levels (due to lack of efficient repression), which in turn will increase unc-3 hypomorph levels.

I think these or similar experiments should be performed to probe the "homeostatic" mechanism, otherwise it might be more accurate to use "robust gene expression levels".

Other minor points:

- Figure 1D: adding lin-11 quantification would be nice as control that cells are there and did not die.
- Is lin-39::AID a hypomorphic allele? it seems even without auxin has some phenotype. If this is the case, it will be good to mention it in the text.
- Figure 2D-E: partial gene expression defects in lin-39 mutants are due to partial penetrance distributed in all locomotion MN or high/total penetrance of phenotype but only for some specific MNs?
- Figure 3: D-E: mab-5 single mutants are missing
- Figure 6-7: for lin-39 locus, do LIN-39 and UNC-3 peaks overlap?, might be interesting to show both in the same figure pannel.

Reviewer #3 (Remarks to the Author):

In the study by Feng et al., the authors characterize the gene regulatory networks involved in the maintenance of cholinergic neuron identity in *C. elegans*. The authors examined the role of two Hox transcription factors, LIN-39 and MAB-5, in regulating cholinergic pathway genes in mature motoneurons. They provide evidence that these factors are required to maintain cholinergic gene expression in the adult nervous system. They also show that the expression of Hox genes is maintained through positive autoregulation, and the levels of Hox expression are negatively regulated by another transcription factor, UNC-3.

In general, this is a nicely done study which adds to the known roles of Hox factors in the regulation of motoneuron fate and connectivity. The major advance of this study is that it provides evidence for a role of Hox genes in more general aspects of neuronal function, specifically the regulation of neurotransmitter synthesis in the adult nervous system. Most previous studies have focused on the roles of Hox factors in generating neuronal subtype-specific features, and more selective functions in gene regulation.

However, the phenotypes of the Hox mutant lines in this study are relatively modest, with single and double Hox mutants showing a somewhat incomplete loss of cholinergic gene expression. Often it appears that only a subset of all the motoneurons are affected (for example the caudal motoneurons shown in Figure 1B for lin39/mab5 mutants). It seems possible that Hox genes are not determining to the maintenance of neurotransmitter identity, as the authors suggest, but are contributing to appropriate levels of gene expression.

In addition it is difficult to interpret some of the results, as in many places the author's refer to changes in gene expression when the quantification shown in the figures reflect the number of motoneurons expressing a given marker or reporter (for example the text reference to Figure 3D). In other cases the authors do quantify the transcript levels and/or level of reporter expression. The authors should refer to data as it is shown in the figures (for example as the fraction of neurons), not with respect to their interpretation about expression.

Another issue relates to the phenotypes of the Hox mutant locomotor neurons. The authors show reduced marker/reporter expression, but we have no idea what is happening to the mutant neurons. Are they still neurons? Are they connected to the correct muscle? In other words are the defects only related to neurotransmitter synthesis. If there are other phenotypes, could the observed changes be indirectly caused by more general aspects of gene expression. This should be addressed for both the

traditional mutant and the auxin-induced models.

The second half of the paper addresses the interactions of Hox genes with themselves and their regulation by the important known cholinergic fate determinant UNC-3. This is a nicely done set of genetic experiments, revealing the complexity of interactions, but am not sure it reveals a novel mechanism of action. It is well known that many TFs autoregulate, and these loops can be modulated by other TFs. Nevertheless this is a nice demonstration of how the network operates in motoneurons.

Manuscript NCOMMS-22-20492

RESPONSE TO REVIEWERS

We thank the reviewers for their time and constructive comments that helped to significantly improve the revised manuscript. We have addressed all raised concerns by conducting new experiments and by modifying the text. A detailed point-by-point response to the reviewers' comments is provided below.

Reviewer #1 (Remarks to the Author):

This study investigates the functional relevance of sustained expression of Hox transcription factors in adult neurons in the ventral cord of *C. elegans*. It was previously observed that neuronal Hox protein expression persists in to adulthood, but its functional significance has not been thoroughly investigated. This paper demonstrates that this extended expression of the Hox transcription factor LIN-39 is necessary for expression of multiple indicators of cholinergic fate in ventral cord neurons that control egg laying and locomotion in *C. elegans* hermaphrodites. The authors additionally reveal a role for the Hox protein MAB-5 and the OE/Collier transcription factor UNC-3 in maintaining cholinergic identity in these neurons. The authors present evidence for direct regulation of cholinergic genes by all three transcription factors, as well as providing evidence for a co-regulatory feedforward loop among the Hox genes and UNC-3.

This work builds on previous studies from the authors' lab demonstrating that UNC-3 interacts with LIN-39 and MAB-5 in its role as a terminal selector for motor neuron identity in *C. elegans*, as well as on a body of previous work that has explored the role of LIN-39 and MAB-5 in specifying male-specific neurons, vulval precursors, and other cell fates in *C. elegans*. A major strength of this paper is that it makes use of reagents that allow temporal control of LIN-39 expression, thus bypassing the previous hurdles presented by the important role of LIN-39 in regulating cell division and survival earlier in larval development. It also makes good use of single-copy endogenous reporters when possible, as well as publicly available ChIP data to make a strong case for the findings presented here.

The work here is solid, significant, and thorough, as it provides mechanistic detail to models hypothesized based on genetic data and expression of multicopy reporter transgenes.

We appreciate the reviewer's supportive comments, highlighting the strengths and significance of this work.

I would like to have seen the authors take more care in acknowledging previous work done on LIN-39 and MAB-5, particularly their post-embryonic roles, and have outlined some examples below:

1) In the introduction (line 111), the statement “whether and how Hox proteins control neurotransmitter identity is unknown,” omits previous work published by Clark et al., 1993 and Salser et al., 1993, demonstrating the requirement for LIN-39 in specifying serotonergic fate of CP neurons in males. I think omitting the “whether” from this statement would resolve this issue.

We thank the reviewer for this suggestion, as it provides an opportunity to make clear in the Introduction what was known before about *lin-39* in the lineage of serotonergic (CP) and cholinergic (VC) neurons. In the revised Introduction, we now cite three papers (Clark et al., 1993; Salser et al., 1993; Loer and Kenyon 1993) on CP lineage and two papers on VC lineage (Potts et al., Dev Bio., 2009; Liu et al., Development, 2006). The paper by Salser states... “*Previous work has shown that the HOM-C gene lin-39 is required for production of these serotonergic motoneurons in the mid-body region (Loer and Kenyon 1993)*”. Similarly, Hunter and Kenyon state in legend of Figure 1: “*In lin-39 mutant males, the P3.aap to P6.aap cells undergo programmed cell death*”.

Prompted by the reviewer’s comment, we have now modified the Introduction as follows:

“Individual neuron types acquire their NT identity during development and maintain it throughout life. Previous work in *C. elegans* showed that the Hox gene *lin-39* (Scr/Dfd/Hox4-5) is required for production of serotonergic neurons (Clark et al., 1993, Salser et al., 1993; Loer and Kenyon, 1993; Hunter and Kenyon, 1995) in males and egg-laying (VC class) motor neurons in hermaphrodites (Potts et al., Dev Bio., 2009; Liu et al., Development, 2006), but whether *lin-39* acts in post-mitotic neurons to regulate NT identity genes was not tested. Hence, the role of Hox proteins in the control of NT identity remains unclear.

2) Also in the male CPs, the work of Hunter and Kenyon (1995) demonstrated that promotion of serotonergic fate can occur as late as the L4 larval stage, suggesting a late/possible ongoing role for LIN-39 in this process as well.

We thank the reviewer for pointing out this experiment. Hunter and Kenyon supplied *lin-39* cDNA with a heat-shock promoter in *lin-39* mutant animals. This elegant rescue experiment, however, was not performed to assess a maintenance role for *lin-39*. For example, the authors should have waited upon heat-shock for the levels of hsp::lin-39 to decrease, and then assess what happens to CPs. Moreover, Table 1 in Hunter and Kenyon states “*The average number of serotonergic CP neurons per rescued animal was 1.4 for heat-shock-lin-39*”. For these reasons, we would prefer to refrain from any claims on a late role of LIN-39 in CP neurons.

3) In the results, the authors mention that Hox proteins can act as developmental timers. Salser and Kenyon (1996) found that MAB-5 seems to play such a role in the male

seam lineage, acting at various stages throughout larval development to control timing of neuronal specification.

We now cite this study in Discussion. Thank you.

The methodology employed is sound and meets the expectations for the field. The authors' combined use of reporters, ChIP, genetics and FISH make a strong case for their findings. They are overall careful in drawing conclusions, and the conclusions follow from the data presented. I have a few specific suggestions about the results section.

1)Line 144 of results: The authors should acknowledge that fosmid reporters are multicopy and do tend to represent overexpression.

We have modified the sentence as follows: *“For the unc-17 and cho-1 expression analysis, we used fosmid (~30kb-long genomic clone)-based reporters, which despite their multicopy nature (overexpression) they tend to faithfully recapitulate endogenous gene expression patterns.”*

2)In the section “LIN-39 and UNC-3 act through distinct binding...” , I found it somewhat confusing that that text states that the regulatory regions of cholinergic genes show (by previously published ChIP) binding to LIN-39, MAB-5, and UNC-3, but that the sites labeled in Figure 4 are only for LIN-39 and UNC-3. It would be helpful to provide clarification in the figure legend for the focus on LIN-39 sites, which I believe was due to the site being well-defined, as stated in the text.

We apologize for this confusion and have modified the figure legend accordingly.

3)Line 291 of results: please change to “A 6.2kb element upstream of the lin-39 locus,” for clarity.

Fixed. Thanks.

4)Results section “Hox genes lin-39 and mab-5 maintain their expression...”. It has been previously shown by Wagmeister et al. (2006) that expression of lin-39 reporters in the ventral cord is regulated by an intronic element. They further showed (via EMSA) that LIN-39 binds to hox consensus sites in this region. Although this paper's finding demonstrates in vivo binding via ChIP, the previous work should be acknowledged.

We regret this oversight. In Results, we now cite twice the Wagmeister et al. (2006) study.

5)Figure 2: the affect in F in adults upon auxin treatment is subtle. It would be helpful to see a representative image for F.

We added a representative image in **Fig. 2G**. The reviewer is right. The effect on *unc-17* expression is subtle because auxin treatment selectively depletes LIN-39, leaving MAB-5 intact to partially compensate. As shown in **Fig. 3D**, the effect on *unc-17* expression is exacerbated in *lin-39(-); mab-5(-)* double mutants compared to *lin-39(-)* single mutants. Regrettably, an auxin-inducible allele for *mab-5* is not available, preventing us from testing a maintenance role for *mab-5*.

6)For all figures showing box and whisker plots for motor neuron number, add means to graphs or figure legends (the dots obscure the mean line in many of these).

We thank the reviewer for this suggestion. In all graphs, we now use a red vertical line to indicate mean value.

Reviewer #2 (Remarks to the Author):

In this work Feng et al. expand on previous work from the group (mainly Feng eLife 2020) to further characterize the gene regulatory networks that establish and maintain motor neuron identity in *C. elegans*.

Previous work from the authors showed that hox LIN-39 and MAB-5 are required to induce and maintain expression of cholinergic Motor neuron (MN) terminal identity genes, this observation is now expanded to include Acetylcholine biosynthesis pathway gene regulation which is described both in locomotor MN and vulva muscle MN. Authors also show that MAB-5 and LIN-39 act directly on the activation of these target genes binding to similar genomic regions as UNC-3 terminal selector, suggesting these factors act together to control MN fate. They also describe that HOX genes activate unc-3 expression, creating a feed forward loop. In addition, HOX also autoregulate their expression. As discussed by the authors, these regulatory motifs are commonly found in cell fate identity networks.

Finally, and most interestingly they also find a repressive action of UNC-3 on lin-39 and mab-5 expression. This described network tightly regulates HOX gene expression levels and this regulation is key for proper MN effector gene expression, as low levels of hox leads to MN gene expression defects but, as authors showed in previous work, high levels of LIN-39 lead to ectopic expression of effector genes from other motoneuron classes. This tight level control is defined by the authors as "homeostatic control". The manuscript is well-written and well-organized, it is easy to follow, experiments are solid and questions are addressed combining several strategies what makes the work very robust. The description of the role of HOX genes in neuron fate maintenance is interesting as well as the importance of tight control of HOX gene expression levels.

My main concern is related to the use of "homeostatic" control to define this mechanism, which I think deserves further work to justify its use in contrast to other terms such as robust gene expression mechanism.

As I understand, homeostatic systems respond to a perturbation with compensatory feedback such that the set point activity of the system is precisely reestablished.

Current data in the paper is very attractive but in my opinion is still not enough to fully support the use of the term homeostatic control.

Indeed, the expectation for homeostatic control would be that ectopic lin-39 expression (performed in Feng 2020 eLife) would lead to increase UNC-3 levels that will in turn decrease expression of lin-39 endogenous locus. This could be experimentally tested using Q-PCR for lin-39 3'UTR to distinguish endogenous gene from lin-39 cDNA overexpression.

Alternatively, unc-3 hypomorphic alleles might lead to an increase of lin-39 expression levels (due to lack of efficient repression), which in turn will increase unc-3 hypomorph levels.

I think these or similar experiments should be performed to probe the "homeostatic" mechanism, otherwise it might be more accurate to use "robust gene expression levels".

We thank the reviewer for this important comment, which prompted us to dig deeper into the literature. As stated in an excellent essay on *Homeostasis* (Billman, 2020, PMID: 32210840):

- a. Homeostasis is defined as “a self-regulating process by which biological systems maintain stability while adjusting to changing external conditions”.
- b. The essay states that Walter Cannon (1871-1945) coined the term homeostasis to mean “staying similar” and *not* “staying the same”.
- c. Homeostasis is the result of complex interactions between negative and positive feedback systems.

We note that our work suggests that ONLY Hox genes (not UNC-3) are under homeostatic control, as mentioned in Abstract, Introduction, Results, and Discussion. We find that tight control of Hox gene (*lin-39*, *mab-5*) expression in *C. elegans* motor neurons is achieved through a two-component mechanism: Hox transcriptional autoregulation (positive feedback) is counterbalanced by negative UNC-3 feedback. Hence, we proposed that these two feedback systems comprise a homeostatic control mechanism for Hox gene expression.

In Discussion, we mention that Hox genes sit at the top of a critical positive feedforward loop (FFL): Hox proteins activate UNC-3. Both Hox and UNC-3 directly activate cholinergic identity genes (e.g., *unc-17/VACHT*, *cho-1/ChT*). Hence, the precise (homeostatic) regulation of Hox gene expression is critical for the functional outcome of this positive FFL – the maintenance of robust expression levels of cholinergic identity genes.

Prompted by the reviewer’s comment, we conducted additional experiments that probe the functional significance of Hox autoregulation and of this FFL.

1. Testing the **functional significance of Hox autoregulation**. In Figure 6, we showed that *lin-39* is required to maintain its own expression in motor neurons. In the revised manuscript, we now provide evidence that disruption of *lin-39* autoregulation results in reduced expression of cholinergic identity genes. Compared to controls, we observed reduced expression of *cho-1/ChT* in motor neurons of animals carrying a *lin-39* allele, in which autoregulation is disrupted (all eleven LIN-39 binding sites in intron 1 of *lin-39* are mutated, **Fig. 6A**). We have now included this new data in **Fig. 6G** and a new sentence in Results.
2. Testing the **functional relevance of the FFL** we describe in **Fig. 7G**. We now show that both arrows from HOX to cholinergic genes and from HOX to UNC-3 are each functionally required. The former is shown in **Fig. 4**; ChIP-Seq data on Hox proteins (LIN-39, MAB-5) and mutational analysis of LIN-39 binding sites demonstrate that these Hox proteins act directly to activate the expression of cholinergic genes (e.g., *unc-17/VACHT*, *cho-1/ChT*). The latter is shown in new **Suppl. Fig. 8**, where we tested what happens to cholinergic genes when we selectively manipulate the binding of LIN-39 to the *unc-3* locus using the *UNC-3::GFP^{LIN-39 site MUT}* allele (shown in **Fig. 5A**). In this allele, the Hox/LIN-39

binding site in the *unc-3* promoter is mutated, resulting in lower levels of UNC-3 protein expression in motor neurons (**Fig. 5C**). Intriguingly, we now find that the expression levels of cholinergic identity genes (*unc-17/VACHT*, *cho-1/ChT*) are increased in motor neurons of animals carrying this allele (**Suppl. Fig. 8**). This increase is consistent with our proposed model where low UNC-3 levels lead to a lack of efficient Hox gene repression, and provides functional relevance for the direct control of *unc-3* by Hox (LIN-39).

3. **Corroborating our initial observations.** In the original version, we found that *lin-39* expression in motor neurons is upregulated in animals carrying a null *unc-3* allele (**Fig. 7**). We have now obtained similar results with a hypomorphic *unc-3* allele (new **Suppl. Fig. 7**, panel A). Similarly, we described in the original version that UNC-3 protein levels are reduced in MNs of animals carrying a null *lin-39* allele (**Fig. 5C**). We have further corroborated these findings with RT-PCR; *unc-3* mRNA levels are reduced in a hypomorphic *lin-39* allele (new **Suppl. Fig. 4**).

Other minor points:

- Figure 1D: adding *lin-11* quantification would be nice as control that cells are there and did not die.

We have added the *lin-11* quantification in a **Suppl. Fig. 1**, new panel **E**. Although there is a small effect in the number of VC neurons expressing *lin-11* in *ced-3(-); lin-39(-)* double mutants, the majority of these animals show *lin-11* expressing in at least 4 of the 6 VC neurons, demonstrating that in *ced-3(-); lin-39(-)* double mutants the majority (if not all) VC neurons are normally generated. On the other hand, no VC neurons express *unc-17/VACHT* in *ced-3(-); lin-39(-)* double mutants (**Fig. 1D**). The small effect on *lin-11* expression is likely a reflection of *lin-11* being a direct LIN-39 target gene, as suggested by our ChIP-Seq data (not shown).

Lastly, the partial effects on two other VC motor neuron markers (*ida-1*, *glr-5*) further indicate that VC neurons are normally generated in *ced-3(-); lin-39(-)* double mutants (**Suppl. Fig. 1D**).

- Is *lin-39::AID* a hypomorphic allele? it seems even without auxin has some phenotype. If this is the case, it will be good to mention it in the text.

Indeed, upon crossing the *lin-39::mNG::3xFLAG::AID* animals with the *eft-3::TIR1* line, we often observe hypomorphic effects in the expression of LIN-39 target genes. This was the case in our previous work (Feng et al., *eLife* 2020, PMID: 31902393) and in the current manuscript. These effects are likely due to a mild reduction in LIN-39 protein levels triggered by TIR1. However, nuclear localization is not affected; LIN-39 protein is present in the nuclei of cholinergic MNs of *lin-39::mNG::3xFLAG::AID; eft-3::TIR1* animals. Despite the hypomorphic effects, we were able to detect statistically significant differences in the expression of LIN-39 target genes (**Fig. 1I**, **Fig. 2F**, **Fig. 6C**).

We now mention that *lin-39::mNG::3xFLAG::AID* is a hypomorphic allele in Results.

Of note, we have observed mild hypomorphic effects when we use the auxin system to deplete other transcription factors (PMID: 28056346). This appears consistent with observations made in other labs using this system in *C. elegans* and beyond.

- Figure 2D-E: partial gene expression defects in *lin-39* mutants are due to partial penetrance distributed in all locomotion MN or high/total penetrance of phenotype but only for some specific MNs?

Leveraging a *cho-1* marker (*cho-1::SL2::mChOpti::H2B*) that localizes *mChOpti* to the nucleus and the fact that the order of MN cell bodies is invariant in the *C. elegans* nerve cord, we identified with single-cell resolution which MNs lack *cho-1* expression. We observe high penetrance of the *lin-39* phenotype, that is, failure to express cholinergic identity genes (e.g., *cho-1/ChT*, *unc-17/VACHT*) in five motor neurons (VA5, VB6, AS6, VA7, VB8) located in the mid-body region. As shown in **Fig. 2A**, these MNs (VA5, VB6, AS6, VA7, VB8) do express the mid-body Hox gene *lin-39*. We have added this new information in the legend of Figure 2. We note that the defects in cholinergic identity gene expression become exacerbated (more MNs fail to express these genes) in *lin-39(-); mab-5(-)* double mutants (**Fig. 3**).

- Figure 3: D-E: *mab-5* single mutants are missing

We have now quantified the effect on *unc-17/VACHT* expression in *mab-5* single mutants (**Fig. 3D**).

- Figure 6-7: for *lin-39* locus, do LIN-39 and UNC-3 peaks overlap?, might be interesting to show both in the same figure panel.

Thanks for this suggestion. The peaks do overlap. We have added ChIP-Seq tracks for UNC-3 and LIN-39 in **Fig. 7A**.

Reviewer #3 (Remarks to the Author):

In the study by Feng et al., the authors characterize the gene regulatory networks involved in the maintenance of cholinergic neuron identity in *C. elegans*. The authors examined the role of two Hox transcription factors, LIN-39 and MAB-5, in regulating cholinergic pathway genes in mature motoneurons. They provide evidence that these factors are required to maintain cholinergic gene expression in the adult nervous system. They also show that the expression of Hox genes is maintained through positive autoregulation, and the levels of Hox expression are negatively regulated by another transcription factor, UNC-3.

In general, this is a nicely done study which adds to the known roles of Hox factors in the regulation of motoneuron fate and connectivity. The major advance of this study is that it provides evidence for a role of Hox genes in more general aspects of neuronal function, specifically the regulation of neurotransmitter synthesis in the adult nervous system. Most previous studies have focused on the roles of Hox factors in generating neuronal subtype-specific features, and more selective functions in gene regulation.

However, the phenotypes of the Hox mutant lines in this study are relatively modest, with single and double Hox mutants showing a somewhat incomplete loss of cholinergic gene expression. Often it appears that only a subset of all the motoneurons are affected (for example the caudal motoneurons shown in Figure 1B for *lin39/mab5* mutants). It seems possible that Hox genes are not determining to the maintenance of neurotransmitter identity, as the authors suggest, but are contributing to appropriate levels of gene expression.

We agree. In **Fig. 3C-E**, we show strong effects on cholinergic identity genes (*cho-1/ChT*, *unc-17/VACHT*, *ace-2/AChE*) in motor neurons of *unc-3* mutants but weaker effects in Hox (*lin-39*; *mab-5*) double mutants. Please, note that there are some gene-specific effects. For example, the loss of *ace-2* expression in **Fig. 3E** is comparable between *unc-3* and *lin-39;mab-5* double mutants. On the other hand, the loss of *unc-17* expression in **Fig. 3D** is striking in *unc-3*, but modest in *lin-39;mab-5* double mutants. These differences in the strength of the phenotypes between *unc-3* and Hox (*lin-39;mab-5*) mutants can be explained by the fact that there must be other factors, in addition to LIN-39 and MAB-5, that contribute to activate *unc-3*. This is based on the fact that in animals carrying strong loss-of-function (null) alleles for both *lin-39* and *mab-5*, we do observe residual *unc-3* expression in motor neurons (**Fig. 5C-D**). To make this important point clear, we added this sentence in Results:

*“We note that *unc-3* expression is not completely abolished in motor neurons of *lin-39*; *mab-5* mutants (**Fig. 5B-I**), indicating that there must be additional, yet-to-be identified factors that activate *unc-3* and compensate for the combined loss of *lin-39* and *mab-5*.”*

We also agree that we often observe stronger effects in caudal motor neurons of Hox double mutants because that is the territory where both *lin-39* and *mab-5* overlap in expression (**Fig. 3A**).

In Response to a similar comment by Reviewer 2, we observe high penetrance of the *lin-39* phenotype, that is, failure to express cholinergic identity genes (e.g., *cho-1/ChT*, *unc-17/VACht*) in five motor neurons (VA5, VB6, AS6, VA7, VB8) located in the mid-body region. We have now added this information in the legend of **Figure 2**. As shown in **Fig. 2A**, these MNs (VA5, VB6, AS6, VA7, VB8) do express the mid-body Hox gene *lin-39*.

We also changed the text in several occasions to indicate that Hox genes contribute to appropriate levels of cholinergic gene expression, instead of claiming that they control cholinergic identity.

In addition, it is difficult to interpret some of the results, as in many places the author's refer to changes in gene expression when the quantification shown in the figures reflect the number of motoneurons expressing a given marker or reporter (for example the text reference to Figure 3D). In other cases, the authors do quantify the transcript levels and/or level of reporter expression. The authors should refer to data as it is shown in the figures (for example as the fraction of neurons), not with respect to their interpretation about expression.

We apologize for the lack of precision in the language used to interpret some of the experiments. We have rectified this issue throughout the revised manuscript and refer to data as shown in figures.

Another issue relates to the phenotypes of the Hox mutant locomotor neurons. The authors show reduced marker/reporter expression, but we have no idea what is happening to the mutant neurons. Are they still neurons? Are they connected to the correct muscle? In other words are the defects only related to neurotransmitter synthesis. If there are other phenotypes, could the observed changes be indirectly caused by more general aspects of gene expression. This should be addressed for both the traditional mutant and the auxin-induced models.

We acknowledge our oversight in describing better the phenotypes of Hox mutant locomotor neurons. In the revised manuscript, we do so based on our findings and previous studies.

In a previous study (Feng et al., 2020, PMID: 31902393), we showed that *lin-39* mutants have locomotion defects. In the current study, we found that *lin-39* is required for the appropriate levels of expression of several cholinergic identity genes in motor neurons (**Fig. 2**). Therefore, we added the following sentence:

“The observed reduction in ACh pathway gene expression may account for the previously described locomotion defects of *lin-39* mutant animals (Feng et al., 2020, PMID: 31902393).”

Moreover, Stefanakis et al., 2015 (PMID: 26291158) found that all motor neurons that control locomotion in the nerve cord of *lin-39(n1760); mab-5(e1239)* double mutant animals normally express markers for synaptic vesicle machinery proteins (RIC-4/Snap25, SNB-1/Vamp/SNARE). Importantly, that paper demonstrated that all these motor neurons are normally generated in *lin-39(n1760); mab-5(e1239)* double mutants. We have modified the text as follows:

“To test this, we generated double *lin-39(n1760); mab-5(e1239)* mutant animals. All MNs that control locomotion are normally generated in these mutants (Stefanakis et al., 2015 (PMID: 26291158). However, we observed a decrease in the expression of the cholinergic marker *unc-17/VACHT* in double *lin-39(n1760); mab-5(e1239)* mutants compared to single *lin-39(n1760)* mutant animals (**Fig. 3D**).”

To test whether cholinergic MNs reach the correct muscle, we crossed a presynaptic GFP marker for cholinergic MNs (*unc-3 promoter::RAB-3::eGFP*) into *lin-39(-); mab-5(-)* double mutant animals. We validated this transgenic marker in a previous study (Kratsios et al., 2015, PMID: 25913400). Next, we quantified the number of presynaptic RAB-3::eGFP positive specialization (boutons) of cholinergic MNs (DA and DB classes) onto the dorsal body wall muscle, which is normally innervated by these neurons. A 200µm-long region was quantified along the dorsal nerve cord, N = 12, L4 stage. Although we found a statistically significant decrease in the number of presynaptic specializations *lin-39(-); mab-5(-)* double mutants (graph on the right), this decrease is driven from the fact that the *unc-3* promoter (used to drive RAB-3::eGFP) is Hox dependent: our current study shows that UNC-3 is a transcriptional target of LIN-39 and MAB-5 (**Fig. 5**). Therefore, it appears that a significant number (if not all) of these cholinergic MNs do reach their appropriate muscle targets in *lin-39(-); mab-5(-)* double mutants based on this quantification of presynaptic boutons.

In addition to the regulation of cholinergic identity genes (this study), Hox genes *lin-39* and *mab-5* are necessary for the expression of five other genes (*del-1/sodium channel*, *unc-77/sodium channel*, *slo-2/potassium channel*, *unc-129/BMP*, *dbl-1/BMP*) (Kratsios

et al., 2017, PMID: 28677525; Feng et al., 2020, PMID: 31902393). These five genes code either for ion channels or signaling molecules and are expressed in specific subtypes (classes) of MNs, not uniformly by all cholinergic MNs as is the case for cholinergic identity determinants (e.g., *unc-17/VACHT*, *cho-1/ChT*). We cited these two papers (Kratsios et al., 2017, PMID: 28677525; Feng et al., 2020, PMID: 31902393) in the original version, lines 453-456.

The cholinergic identity determinants (e.g., *unc-17/VACHT*, *cho-1/ChT*) are direct targets of Hox (LIN-39, MAB-5) (this study). The same appears to be the case for the aforementioned five genes (Feng et al., 2020, PMID: 31902393).

To summarize, the motor neurons that control locomotion in Hox (*lin-39*; *mab-5*) mutants (a) are normally generated, (b) the majority of them (if not all) reach their muscle targets, (c) they show reduced levels of cholinergic identity gene expression (e.g., *unc-17/VACHT*, *cho-1/ChT*), and (d) they also show reduced levels in the expression of genes encoding ion channels and other signaling molecules (*unc-129*, *del-1*, *unc-77*, *dbl-1*, *slo-2*). It is unlikely that the observed changes in cholinergic identity genes (e.g., *unc-17/VACHT*, *cho-1/ChT*) in motor neurons of Hox mutants are indirectly caused by more general changes in gene expression for two reasons.

First, the cholinergic identity genes (e.g., *unc-17/VACHT*, *cho-1/ChT*) are direct targets of Hox (LIN-39, MAB-5) proteins, as show in **Fig. 4**. Moreover, UNC-3 is also a direct Hox target (**Fig. 5**). Hence, the reduction of cholinergic identity genes in Hox mutants is due to their direct (via Hox) and indirect (Hox → UNC-3) regulation.

Second, we showed in a previous study that, in *C. elegans* motor neurons, the expression of the cholinergic identity determinant UNC-3 and the structure of cholinergic neuromuscular synapses are both unaffected when global neurotransmission was genetically blocked (Kratsios et al., 2015, PMID: 25913400), suggesting the expression of cholinergic identity genes is independent of neuronal activity in *C. elegans* motor neurons.

The second half of the paper addresses the interactions of Hox genes with themselves and their regulation by the important known cholinergic fate determinant UNC-3. This is a nicely done set of genetic experiments, revealing the complexity of interactions, but am not sure it reveals a novel mechanism of action. It is well known that many TFs autoregulate, and these loops can be modulated by other TFs. Nevertheless, this is a nice demonstration of how the network operates in motoneurons.

We do agree with the reviewer that feed-forward and autoregulatory loops have been described before in many biological systems and we acknowledge this in several places in Discussion. The novel aspects of the second part of our study are:

- We show that Hox autoregulation in non-dividing, long-lived cells (motor neurons) is continuous and required during post-embryonic life to maintain the right levels of cholinergic identity gene expression.
- We have not identified any studies in the literature describing a homeostatic mechanism that ensures continuous/robust expression of genes required for neuronal function (e.g., NT identity genes).

REVIEWERS' COMMENTS

Reviewer #1 (Remarks to the Author):

The authors have thoroughly and thoughtfully addressed my previous concerns in this resubmitted manuscript.

Reviewer #2 (Remarks to the Author):

The revised version of the Manuscript by Feng et al. has addressed all my comments and question, including new experimental data and clarifications where needed.

Reviewer #3 (Remarks to the Author):

The authors have addressed my major concerns as well as the concerns of the other reviewers in their revision. They have performed additional experiments to address the fate of the neurons in Hox mutants, and have clarified their role in maintaining appropriate levels of cholinergic gene expression.

RESPONSE TO REVIEWERS

Reviewer #1 (Remarks to the Author):

The authors have thoroughly and thoughtfully addressed my previous concerns in this resubmitted manuscript.

Reviewer #2 (Remarks to the Author):

The revised version of the Manuscript by Feng et al. has addressed all my comments and question, including new experimental data and clarifications where needed.

Reviewer #3 (Remarks to the Author):

The authors have addressed my major concerns as well as the concerns of the other reviewers in their revision. They have performed additional experiments to address the fate of the neurons in Hox mutants, and have clarified their role in maintaining appropriate levels of cholinergic gene expression.

We are glad that all three reviewers agree that the revised manuscript has addressed their concerns.